# Identification of two major autoantigens negatively regulating endothelial activation in Takayasu arteritis

Tomoyuki Mutoh [1,6], Tsuyoshi Shirai [1,6 ✉], Tomonori Ishii[1], Yuko Shirota [1], Fumiyoshi Fujishima[2], Fumiaki Takahashi [3], Yoichi Kakuta [4], Yoshitake Kanazawa[4], Atsushi Masamune [4], Yoshikatsu Saiki[5], Hideo Harigae[1] & Hiroshi Fujii[1]

The presence of antiendothelial cell antibodies (AECAs) has been documented in Takayasu arteritis (TAK), a chronic granulomatous vasculitis. Here, we identify cell-surface autoantigens using an expression cloning system. A cDNA library of endothelial cells is retrovirally transfected into a rat myeloma cell line from which AECA-positive clones are sorted with flow cytometry. Four distinct AECA-positive clones are isolated, and endothelial protein C receptor (EPCR) and scavenger receptor class B type 1 (SR-BI) are identified as endothelial autoantigens. Autoantibodies against EPCR and SR-BI are detected in 34.6% and 36.5% of cases, respectively, with minimal overlap (3.8%). Autoantibodies against EPCR are also detected in ulcerative colitis, the frequent comorbidity of TAK. In mechanistic studies, EPCR and SR-BI function as negative regulators of endothelial activation. EPCR has also an effect on human T cells and impair Th17 differentiation. Autoantibodies against EPCR and SR-BI block the functions of their targets, thereby promoting pro-inflammatory phenotype.

[1] Department of Hematology and Rheumatology, Tohoku University Graduate School of Medicine, Sendai, Japan. [2] Department of Anatomic Pathology, Tohoku University Graduate School of Medicine, Sendai, Japan. [3] Division of Medical Engineering, Department of Information Science, Iwate Medical University, Morioka, Japan. [4] Division of Gastroenterology, Tohoku University Graduate School of Medicine, Sendai, Japan. [5] Division of Cardiovascular Surgery, Tohoku University Graduate School of Medicine, Sendai, Japan. [6]These authors contributed equally: Tomoyuki Mutoh, Tsuyoshi Shirai. ✉email: tsuyoshirajp@med.tohoku.ac.jp

Takayasu arteritis (TAK), a type of large vessel vasculitis (LVV), affects the aorta and its major branches[1]. Because there does not exist an animal model, TAK studies have been conducted using human samples; its pathogenesis is mostly unclear[2,3]. Granulomatous vasculitis is a typical pathological finding, and T cells have been implicated as key players; myeloid cells, including macrophages, are effector cells that promote disease progression[4,5]. In addition to such cellular immune reactions, the involvement of B cells and autoantibodies has been described[6]. Initially, the presence of antiaortic antibodies was documented by complement fixation test and hemagglutination test with homogenized human aorta[7,8]. Since the 1990s, high numbers of patients with TAK who had antiendothelial cell antibodies (AECAs) have been reported[9–11]; several studies identified the pathogenic effects of AECAs in TAK, including endothelial cell activation[12], cytotoxicity[13], cytokine production, and apoptosis[14]. Recently, IgG from patients with TAK was found to activate the mammalian target of rapamycin pathway[15].

Identifying target antigens has been difficult. Target antigens of AECAs are heterogeneous and include membrane component, ligand-receptor complex, and molecule adhering to plasma membrane[16]. Autoantigens may be either constitutively expressed and translocated from intracellular compartment to membrane. Anti-neutrophil cytoplasmic antibodies are one of the auto-antibodies which recognize intracellular antigens and their pathogenic roles have been implicated[17]. Immunoprecipitation and proteomics using two-dimensional electrophoresis are pre-ferred for autoantigen identification[18]. These detection methods do not differentiate between cell-surface molecules and intracel-lular molecules. Furthermore, extraction of some membrane proteins is difficult in proteomics analysis, making it challenging to identify membrane proteins like AECA targets[19]. Therefore, we constructed an expression cloning system to identify cell-surface antigens: serological identification system for autoantigens using a retroviral vector and flow cytometry (SARF)[18,20]. In SARF, a cDNA library of endothelial cells is integrated into mammalian cells by retroviral vectors. Cells expressing the cDNA library are stained with prototype AECAs and fluorescent-conjugated secondary anti-human IgG, and cells with fluorescence are sorted. Autoantigen identification is performed by analyzing the cDNA inserted into the sorted cells. Our studies have shown the usefulness of this system for identifying cell-surface autoantigens[18,20,21].

One of the greatest issues in clinical practice is the absence of disease-specific testing for TAK[22]. Relapse is frequent, which accounts for approximately 60% of patients[23]. Although inflammatory markers and imaging studies are now used for TAK diagnosis and management, interpretation of these results is sometimes difficult because of their non-specificity. Therefore, disease-specific autoantibodies in TAK would be extremely important for clinical application and elucidating pathophysiology.

In this study, we therefore aim to identify autoantigens in TAK by using SARF, and successfully identify two membrane proteins, endothelial protein C receptor (EPCR) and scavenger receptor class B type 1 (SR-BI), as major autoantigens in TAK.

## Results

### Sorting of cells expressing cell-surface autoantigens for TAK.
AECA activities for human aortic endothelial cells (HAECs) and human umbilical vein endothelial cells (HUVECs) treated with or without tumor necrosis factor-α (TNF-α) were compared in 21 patients with TAK (Supplementary Fig. 1); no significant differ-ences were noted (Fig. 1a). Therefore, we generated a cDNA library from unstimulated HUVECs.

Next, we selected nine AECA serum samples for SARF and successfully completed SARF using three serum samples (U10-4, W10-59, and G10-43) whose AECA activities are shown in Fig. 1b. Using SARF, HUVEC cDNA–expressing YB2/0 cells were incubated with U10-4 IgG and FITC-conjugated secondary antibody, and cells with strong FITC signals were sorted (Fig. 1c). After cell expansion, we performed two more rounds of cell sorting and found cells with strong autofluorescence in the sorted cells. Therefore, PE-conjugated antibody was used instead of the FITC-conjugated secondary antibody, and cells with strong PE signals were further sorted. After the sixth round of sorting, cells bound to U10-4 IgG markedly increased. Then, two clones (C1 and C3) were established from the U10-4 IgG-binding cell population by the limiting dilution method (Fig. 1d). Similarly, SARF was conducted with W10-59 IgG (Fig. 1e, f) and G10-43 IgG (Fig. 1g, h), and one clone was isolated from each serum sample (C6 from W10-59 and C7 from G10-43; Fig. 1f, h).

### EPCR and SR-BI as cell-surface autoantigens in TAK.
After PCR amplification and cloning of HUVEC cDNA inserted into the genomic DNA of C1 and C3 clones from the U10-4 sample (Fig. 2a), DNA sequencing and BLAST analysis were performed. PCR bands from two clones at around 2000 bp were found to correspond to the same gene PROCR (GenBank accession num-ber NM 006404.4, Fig. 2b) encoding EPCR, and EPCR expression on the cell surfaces of the C1 and C3 clones was confirmed (Fig. 2c). Next, we generated EPCR-expressing YB2/0 cells (Supplementary Fig. 2a–c). U10-4 IgG showed significant binding activity to EPCR-expressing cells (Fig. 2d). Incubation with soluble recombinant EPCR protein inhibited this U10-4 IgG binding (Fig. 2e). In addition, the binding activity of U10-4 serum to recombinant EPCR protein was confirmed by Western blotting (Fig. 2f).

PCR amplification of C6 isolated from W10-59 and C7 isolated from G10-43 showed similar bands around 3000 bp (Fig. 2g). DNA sequencing revealed that these bands corresponded to the same gene SCARB1 (GenBank accession number NM 005505.4, Fig. 2h) encoding SR-BI. Similar to anti-EPCR activity in the U10-4 sample, anti-SR-BI activity was confirmed in the W10-59 and G10-43 samples (Fig. 2i–l, Supplementary Fig. 2d–f). Importantly, anti-SR-BI activity of AECA was not documented by the standard Western blotting. However, we confirmed anti-SR-BI activity by using immunoprecipitation (Fig. 2l), indicating that the spatial structure of SR-BI protein was important for the binding of anti-SR-BI autoantibodies. To further confirm cDNA inserted into cloned cells, we performed microarray analysis to compare expressions of mRNA between cloned cells and untransfected YB2/0 cells. Microarray analysis showed that EPCR and SR-BI signals dramatically increased in cloned cells (Supplementary Fig. 3a). Anti-EPCR and anti-SR-BI activities were examined in nine prototype TAK serum samples for SARF. Anti-EPCR or anti-SR-BI activity was observed in four serum samples each, without overlap (Supplementary Fig. 3b).

### Expression of EPCR and SR-BI in TAK tissue.
Although thickening of intimal layers of the aorta is the hallmark of TAK, this has been considered as the secondary phenomenon and the main inflammatory site of TAK is in vasa vasorum[24]. To inves-tigate the expression of autoantigens in TAK, immunohis-tochemistry was performed using resected aorta from TAK patients. Vasa vasorum vasculitis with infiltration of inflamma-tory cells was observed in TAK tissue, and endothelium of vasa vasorum expressed both EPCR and SR-BI (Fig. 3). Although EPCR and SR-BI were also expressed in the endothelial cells of the affected aortic lumen (Supplementary Fig. 4), their

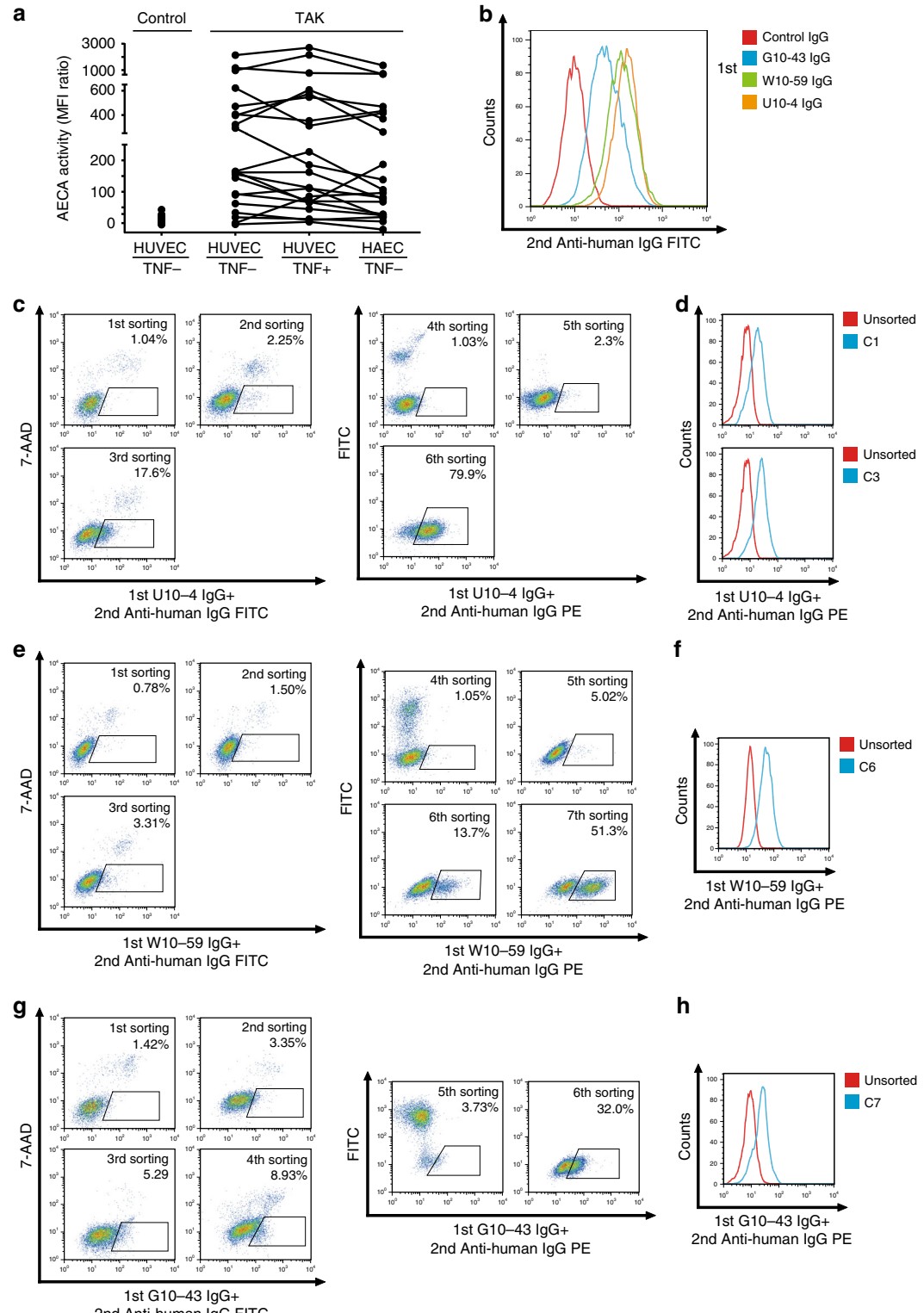

**Fig. 1 Subcloning of autoantigen-expressing cells by using IgG from patients with Takayasu arteritis. a** AECA activity against HAECs or HUVECs treated with or without 1 ng/mL TNF-α for 6 h was measured with flow cytometry in patients with Takayasu arteritis (TAK, n = 21). Dots represent the data for individual subjects. Dots connected with lines indicate the same patients. Control indicates healthy individuals (n = 79). **b** Nonpermeabilized HUVECs were stained with 0.5 mg/mL purified IgG obtained from a healthy individual or patients with TAK, followed by incubation with secondary antibody and flow cytometry analysis. **c**–**h** Sorting of cells expressing cell-surface autoantigens by using three different AECAs. **c**, **e**, **g** YB2/0 cells expressing HUVEC cDNA were stained with 0.5 mg/mL U10-4 IgG (**c**), W10-59 IgG (**e**), or G10-43 IgG (**g**), followed by incubation with secondary antibody, and cells in the positive fraction (squares) were sorted by flow cytometry. FITC-conjugated IgG antibody was initially used, and PE-conjugated IgG antibody was subsequently used as a secondary antibody. **d**, **f**, **h** Binding activity of serum IgG to unsorted cells or cloned cells from sorted cells (C1 [upper] and C3 [lower] isolated from U10-4, D; C6 isolated from W10-59, F; C7 isolated from G10-43, H, respectively); cells were stained with 0.5 mg/mL prototype AECA IgG, followed by incubation with secondary antibody and flow cytometry analysis. G10-43, W10-59, and U10-4 indicate the serum sample number.

expressions were more intense in the vasa vasorum, suggesting their roles in vasa vasorum vasculitis. Immunohistochemistry was also performed in non-inflammatory aortic tissue to investigate expressions of EPCR and SR-BI as disease controls. Surgical specimen from patients with aortic aneurysm and aortic stenosis, and an autopsy sample from a patient with hypothyroidism were analyzed. The expression of EPCR was not evident in the intima, and the endothelium of vasa vasorum was stained weakly in non-inflammatory aortic tissue (Fig. 3a and Supplementary Fig. 4a). The expression of SR-BI was also detected in the endothelium of vasa vasorum, whereas the intimal layer was not stained in some samples of non-inflammatory aortic tissue (Fig. 3b and Supplementary Fig. 4b).

**Distribution of patients with anti-EPCR or anti-SR-BI activity.** To validate these autoantibodies in TAK, we measured anti-EPCR and anti-SR-BI activities in 52 patients with active TAK (Fig. 4a, b and Supplementary Fig. 1). Anti-EPCR antibodies were detected in 18 patients (34.6%), and anti-SR-BI antibodies, in 19 (36.5%); both antibodies were detected in two patients (Fig. 4c). The anti-EPCR or anti-SR-BI activity of these serum samples was confirmed by inhibition tests (Supplementary Fig. 5).

To test autoantibody specificity for TAK, anti-EPCR, and anti-SR-BI activities were also measured in other vasculitides and autoimmune rheumatic diseases (Fig. 4a, b). Ten patients who had giant cell arteritis (GCA) with positive results in temporal artery biopsy possessed neither of them although three patients manifested large-vessel involvement. Anti-EPCR activity was detected in 1 of 18 patients with Sjögren's syndrome and 4 of 93 patients with systemic lupus erythematosus (SLE); anti-SR-BI activity was detected in 1 of 14 patients with microscopic polyangiitis and 4 of 93 patients with SLE. In summary, the sensitivity and specificity of two auto-antibodies in autoimmune rheumatic diseases were 67.3 and 98.0%, respectively.

**Clinical characteristics of patients with TAK who had autoantibodies.** The coexistence of both autoantibodies was rare ($n = 2$); most patients with active TAK were classified as anti-SR-BI single positive ($n = 17$), anti-EPCR single positive ($n = 16$), or double negative ($n = 17$). Their clinical characteristics have been presented in Supplementary Tables 1, 2. We selected ten parameters which we considered as clinically important and performed statistical analysis as shown in Table 1. To correct multiplicity, $P < 0.005$ was considered to be statistically significant using Bonferroni method.

Anti-SR-BI-positive patients were relatively older (mean, 41.2 years), and aortic regurgitation (AR) was relatively less than in other types. They exhibited elevated levels of inflammatory markers, including C-reactive protein ($P = 0.004$, one-way ANOVA), and relative elevation of erythrocyte sedimentation rate; 64.7% patients had type V artery lesions. Patients with anti-EPCR autoantibodies tended to experience more strokes (25.0%) and had significantly higher frequencies of ulcerative colitis (UC) ($P = 0.004$, $\chi^2$ test). Lesser numbers of arteries were affected, and 62.5% patients had type II artery lesions. Patients without these autoantibodies had increased rates of surgery (41.2%), most of which were performed for AR.

Serial measurements of these autoantibodies showed that anti-EPCR autoantibody titers decreased following immunosuppressive treatment and anti-SR-BI autoantibodies increased on relapse (Supplementary Fig. 6).

Because the complication of UC was significantly higher in patients with anti-EPCR autoantibodies, we measured anti-EPCR autoantibodies in 35 patients with primary UC. Surprisingly, 68.6% of UC sera possessed binding activities to EPCR (Table 2 and Supplementary Fig. 7), suggesting common pathophysiology among TAK and UC. Anti-SR-BI autoantibodies were not detected in primary UC.

**Anti-EPCR and SR-BI autoantibodies increased endothelial activation.** EPCR is a receptor for activated protein C (APC), and binding of APC to EPCR results in anti-inflammatory activity[25–27]. TNF-α (pro-inflammatory cytokine) activated the endothelium and upregulated adhesion molecules, including E-selectin, vascular cell adhesion molecule 1, and intercellular adhesion molecule 1 (Fig. 5a). APC addition inhibited upregulation of adhesion molecules by TNF-α (Fig. 5a); this effect was noted at the transcriptional level (Fig. 5b). Pre-incubation of HUVECs with anti-EPCR antibody or AECA IgG with anti-EPCR activity (J11-14) at the concentration corresponding to 1:10 diluted serum blocked the protective effect of APC on adhesion molecules (Fig. 5c, d), whereas incubation with control IgG or AECA-negative TAK IgG did not. In the absence of APC, pre-incubation with J11-14 IgG did not change expressions of adhesion molecules (Supplementary Fig. 8a). This blocking effect of IgG was dose dependent (Supplementary Fig. 8b); recombinant EPCR proteins reversed the blocking effect (Supplementary Fig. 9). Anti-EPCR autoantibodies from different patients with TAK also showed blocking activity (Supplementary Fig. 10). Other endothelial cells including HAECs and human pulmonary endothelial cells (HPAECs) also expressed EPCR, and similar blocking effect of anti-EPCR autoantibodies was also observed in these cells (Supplementary Fig. 11).

One of the ligands for SR-BI is high density lipoprotein (HDL)[28–30], which also acts as a negative regulator of endothelial activation (Fig. 5e). Anti-SR-BI-positive TAK AECAs (M11-36) inhibited HDL uptake in HUVECs (Supplementary Fig. 12a, b). Anti-EPCR-positive TAK AECAs did not interfere the uptake of HDL (Supplementary Fig. 12c). Commercially available anti-SR-BI IgG and anti-SR-BI-positive TAK AECAs also dose-dependently blocked the protective effect of HDL on TNF-α-treated HUVECs (Fig. 5f and Supplementary Figs. 13, 14), which was reversed by the addition of recombinant SR-BI (Supplementary Fig. 15). HAECs and HPAECs also expressed SR-BI, and anti-SR-BI autoantibodies blocked the effects of HDL (Supplementary Fig. 16). Thus, anti-SR-BI autoantibodies also promoted endothelial activation. HDL exerts anti-inflammatory actions mainly by increasing nitric oxide bioavailability[31]. Measurement of nitric oxide synthase (NOS) activity revealed that anti-SR-BI autoantibodies suppressed NOS activity (Supplementary Fig. 17).

**Anti-EPCR autoantibodies promoted Th17 differentiation.** T cells are the main players in TAK[2], and association with Th17 cells has been implicated[32,33]. Recently, Kishi et al. reported that EPCR negatively regulated Th17 differentiation in mice[34]. Therefore, we hypothesized that anti-EPCR autoantibodies could alter Th17 differentiation in human TAK.

Human naive CD4 T cells were stimulated with CD3/CD28 beads, and EPCR expression was analyzed. EPCR expression was induced under Th17 differentiation conditions, and approximately 30% cells expressed EPCR on Day 7 (Fig. 6a, b). APC addition restricted Th17 differentiation; this phenomenon was reversed by adding commercially available anti-EPCR IgG (Fig. 6c). To confirm the effect of anti-EPCR antibody from TAK AECAs (J11-14), CD4 T cells were cultured with control IgG or J11-14 IgG. J11-14 IgG disturbed the negative effect of APC on Th17 differentiation and promoted

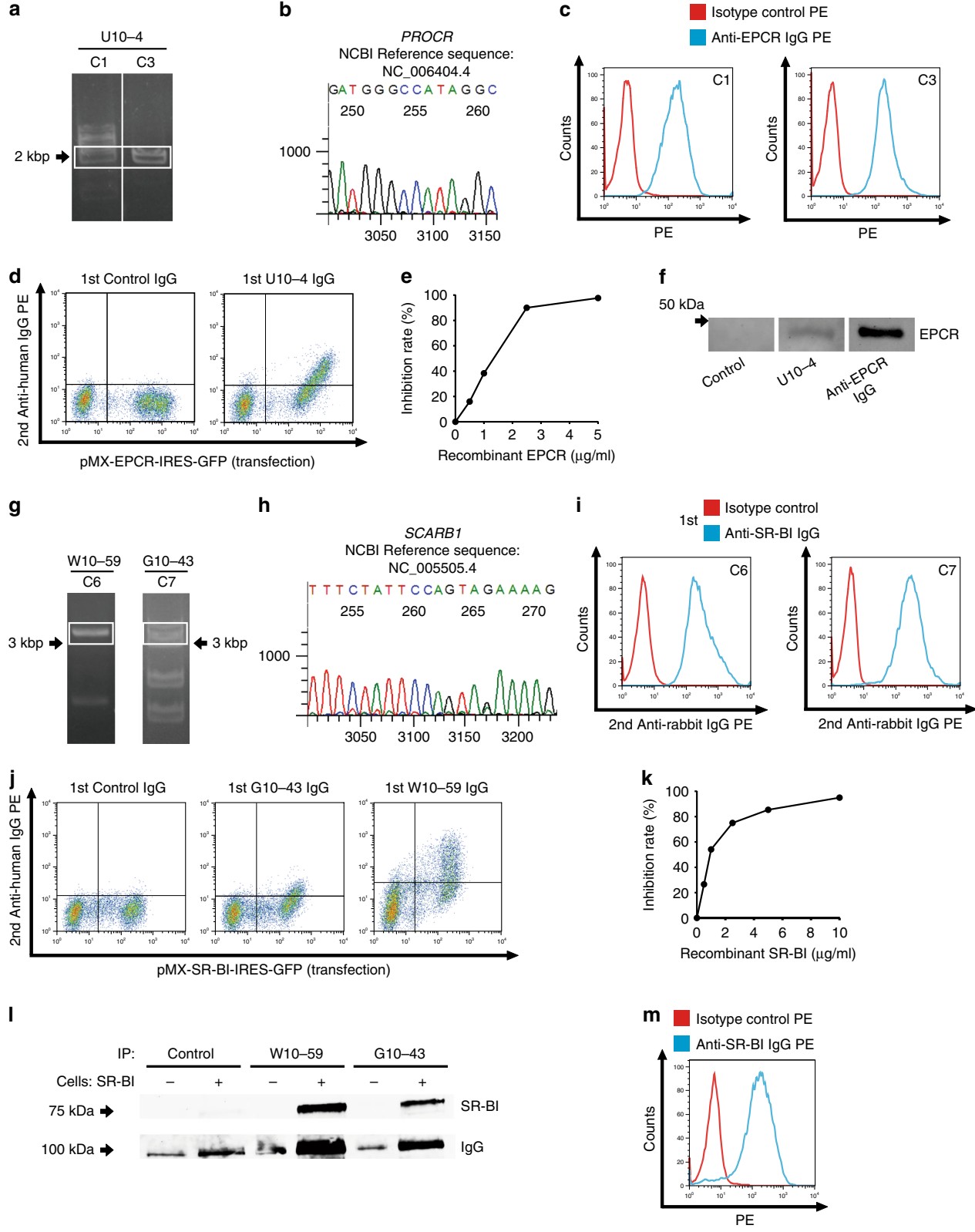

Th17 differentiation (Fig. 6d). This effect was further confirmed at the transcriptional level. Anti-EPCR autoantibodies blocked retinoid acid receptor-related orphan receptor γt, interleukin (IL)-17A, IL-17F, IL-21, and IL-22 downregulation (Fig. 6e).

## Discussion

Identification of major autoantigens in TAK would significantly affect clinical practice and vasculitis research. In this study, we analyzed autoantigens in TAK by using SARF. EPCR and SR-BI (integrated membrane protein receptors) were found to be major

**Fig. 2 Identification of endothelial protein C receptor (EPCR) and scavenger receptor class B type 1 (SR-BI) as endothelial autoantigens in Takayasu arteritis. a** HUVEC cDNA fragments inserted into the genomic DNA of C1 and C3 clones established with U10-4 IgG were amplified, and PCR products were electrophoresed on a 0.8% agarose gel. **b** DNA sequencing was performed for the PCR products obtained around 2000 bp for C1, followed by BLAST analysis. **c** C1 (left) and C3 (right) were stained with PE-conjugated isotype control or PE-conjugated anti-human EPCR antibody and analyzed with flow cytometry. **d** The expression vector EPCR-IRES-GFP was transfected into YB 2/0 cells, and the cells were stained with 0.5 mg/mL control IgG or U10-4 IgG, followed by incubation with secondary antibody and flow cytometry analysis. **e** Inhibition tests for binding activities to YB2/0 cells overexpressing EPCR were performed using 0.5 mg/mL U10-4 IgG with soluble recombinant EPCR at the indicated concentrations. **f** Western blotting of recombinant EPCR proteins was performed, and they were stained with control serum, U10-4 serum, or anti-human EPCR antibody, followed by secondary antibodies. **g** HUVEC cDNA fragments inserted into the genomic DNA of C6 clones established with W10-59 IgG and C7 by using G10-43 IgG were amplified, and PCR products were electrophoresed on a 0.8% agarose gel. **h** DNA sequencing was performed for the PCR products obtained around 3000 bp for C7, followed by BLAST analysis. **i** C6 (left) and C7 (right) were stained with anti-human SR-BI antibody or isotype control, followed by incubation with secondary antibody and flow cytometry analysis. **j** The expression vector SR-BI-IRES-GFP was transfected into YB 2/0 cells, and the cells were stained with 0.5 mg/mL control IgG, W10-59 IgG, or G10-43 IgG, followed by incubation with secondary antibody and flow cytometry analysis. **k** Inhibition tests for binding activities to YB2/0 cells overexpressing SR-BI were conducted using 0.5 mg/mL W10-59 IgG with soluble recombinant SR-BI at the indicated concentrations. **l** YB2/0 cells expressing with or without SR-BI were reacted with control, W10-59, or G10-43 serum. Cells were then lysed and immunoprecipitation was performed. Western blotting was then performed, and the membrane was analyzed for the expression of SR-BI and human IgG. **m** Nonpermeabilized HUVECs were stained with PE-conjugated anti-human SR-BI antibody or isotype control and analyzed with flow cytometry. G10-43, W10-59, and U10-4 indicate the serum sample number.

autoantigens in TAK and were intensely expressed in the vasa vasorum of TAK tissue. The EPCR induces antithrombotic activity, cytoprotective activity, and barrier protective effects[25]. SR-BI is involved in cholesterol homeostasis and inhibits endothelial inflammation and cell death. Because membrane protein identification via proteomic analysis is difficult[19], the current success would be attributable to SARF. Meanwhile, endothelial cells are naturally adherent and polarized cells. Cells should be detached from the plate for detection by flow cytometry, and this may lead to change and internalization of surface antigens. Taken together, use of SARF for autoantigen identification could be considered for other diseases in which AECAs have been reported[18].

Validation of autoantibodies in 52 patients with active TAK, other vasculitides, and other autoimmune rheumatic diseases showed their high specificity in TAK among autoimmune rheumatic diseases (sensitivity, 67.3%; specificity, 98.0%). These data suggested the potential diagnostic role of these autoantibodies. Interestingly, none of GCA patients in this study possessed these autoantibodies, suggesting different etiology among TAK and GCA. Each autoantibody was found in about one-third of the patients; overlap was observed only in two cases. In a previous study, autoantibodies against EPCR were measured in patients with antiphospholipid syndrome (APS)[35]. EPCR was identified as an autoantigen by global screening of HUVEC cDNA in the current study, and high anti-EPCR activity was not detected in patients with SLE complicated by APS in our cohort. Our method detected extracellular domain of EPCR in live cells, while they utilized enzyme-linked immunosorbent assay and measured binding activity to recombinant EPCR proteins. This difference might be important for the specificity of anti-EPCR autoantibodies in TAK. Because high titer of these autoantibodies was not detected in other autoimmune rheumatic diseases, anti-EPCR and anti-SR-BI autoantibodies were considered fairly specific for TAK among autoimmune rheumatic diseases. On the other hand, we further revealed that patients with primary UC possessed autoantibodies against EPCR. Crucial role of EPCR in governing microvascular inflammation in inflammatory bowel disease has been reported[36]. This data in itself has significant impact on the research for UC because of its high prevalence. The co-existence of TAK and UC has been known, and they further share common complications such as spondyloarthritis and pyoderma gangrenosum[37]. Therefore, this result suggests the similar pathogenesis among TAK and UC, which can be characterized by the presence of anti-EPCR autoantibodies. Further studies are required to evaluate association among TAK and UC based on this autoantibody.

Our results suggest three TAK subclasses on the basis of presence of autoantibodies: anti-SR-BI positive, anti-EPCR positive, or double negative. Analysis of the clinical features of the groups revealed relatively distinct phenotypes. In our cohort, patients with anti-SR-BI autoantibodies had wider distribution of LVV and higher inflammatory marker levels than other subgroups and relatively late onset; more than 60% had type V artery lesions. Despite higher inflammatory marker levels, AR frequency was lesser in this group. In anti-EPCR-positive patients, the incidence of cerebrovascular events was higher than in other subgroups and a significant association with UC was observed. Furthermore, 62.5% patients had only type II artery lesions, and abdominal aorta involvement was less common. However, this data do not indicate that the disease activity of these patients was low because inflammatory marker levels are less reliable for determining disease activity in TAK[38]. Anti-EPCR-positive and double-negative patients had greater frequency of AR than the anti-SR-BI-antibody group. These results suggest intense vascular inflammation in the ascending aorta in the first two groups. The surgical intervention tended to be the highest in the double negative group. These data suggest the potential use of these autoantibodies for the subclassification of TAK. Furthermore, the data from serial measurements implicate that these autoantibodies reflect disease activity, suggesting their roles for monitoring disease activity. Taken together, these autoantibodies could be used as a diagnostic, subclassification, and monitoring tool in clinical practice. Because disease-specific markers are extremely helpful, further validation in a larger population should be conducted.

Identification of disease-specific autoantigens improves the understanding of TAK pathogenesis. Autoantigen identification supports the presence of autoantigen-specific B cells in TAK, which require aid from T cells. In the current study, we further revealed the pathogenic potential of these autoantibodies (Fig. 7). Because the autoantigens identified were plasma membrane protein receptors, they had the potential to either block or stimulate receptor signaling. We showed that anti-EPCR and anti-SR-BI autoantibodies functioned as blocking antibodies (Figs. 5, 6). Because their targets are negative regulators of inflammation, the autoantibodies disturb their anti-inflammatory activities, thereby promoting vascular inflammation in vasa vasorum. Particularly, it is possible that suppression of NOS activation underlid the inhibitory action of anti-SR-BI autoantibodies. In humans, a loss-of-function variant of SR-BI was reported to be correlated with increased levels of HDL, and an increased risk of coronary heart disease[39]. In our cohort, there did not exist significant difference in the lipid profiles depending on the presence

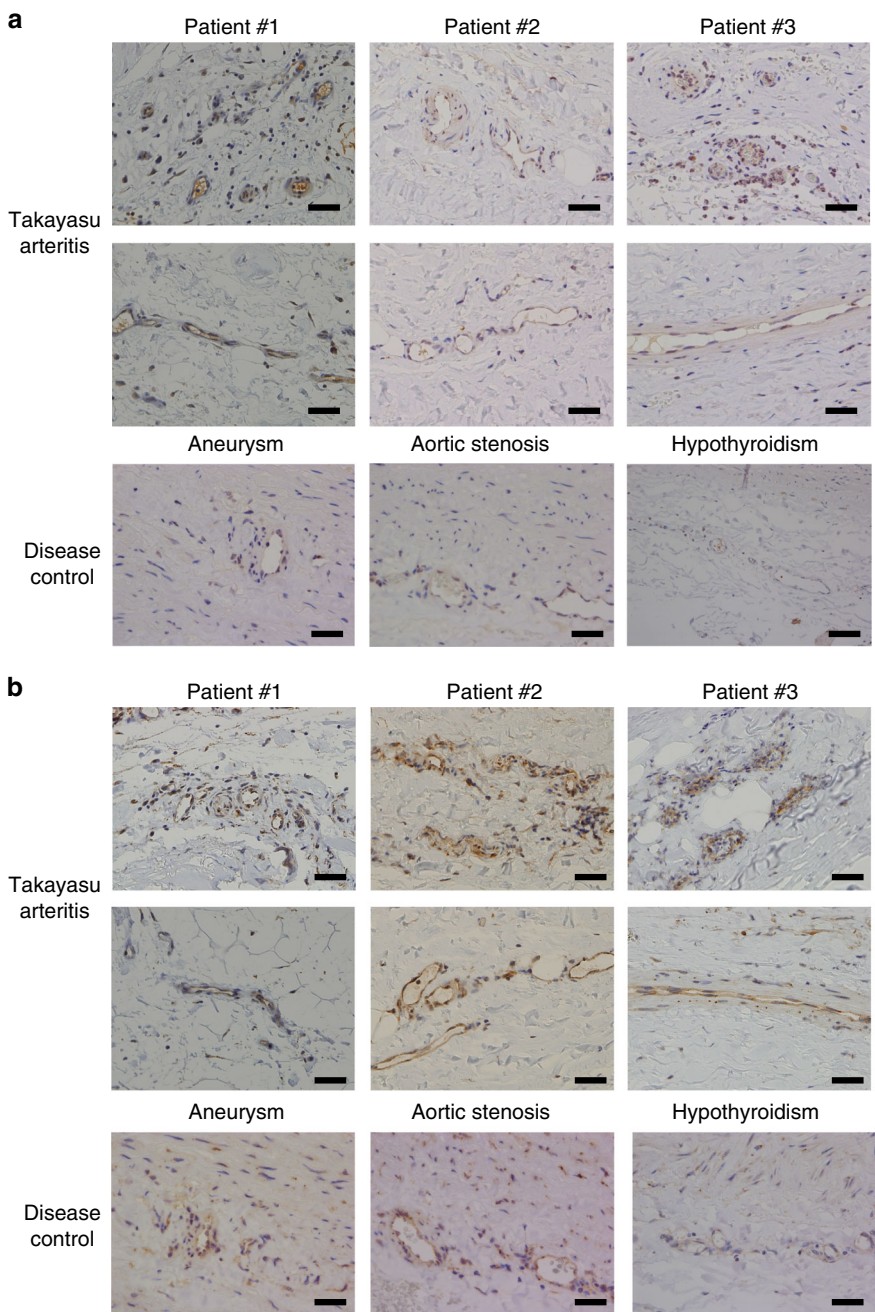

**Fig. 3 Expressions of EPCR and SR-BI in the vasa vasorum of the aorta.** Immunohistochemical analysis was performed against EPCR (**a**) and SR-BI (**b**) in resected aortic tissue from three different patients with TAK (upper two columns) and non-inflammatory aortic tissue as disease controls (lower column). Disease controls included surgical specimen from patients with aortic aneurysm and aortic stenosis, and an autopsy sample from a patient with hypothyroidism. Expressions in the vasa vasorum of the aorta are shown. Black bar indicates 40 μm.

of autoantibodies. It is possible that lipid profiles were affected by the higher inflammatory activities of anti-SR-BI positive subjects. However, the inhibition of SR-BI by autoantibodies could be the risk for developing atherosclerotic lesions whose occurrence is known to be high in TAK[40]. Further investigations of these autoantibodies in severe atherosclerosis and thrombotic disease would also be important.

We found that anti-EPCR antibodies blocked the negative effect of APC on Th17 differentiation, thus promoting this differentiation. This is a remarkable finding because interference or promotion of Th17 differentiation by autoantibodies has not been previously reported. Considering the pathogenesis of TAK[41], this function seems quite reasonable and would be

important because aberrant Th17 activation has been documented in TAK.

Because both autoantibodies seemed to function in disturbing negative regulation and promoting vascular inflammation, their generation might be the consequence rather than the initiator of vascular inflammation. One hypothesis is that vascular damage accompanied by immune activation initiates and leads to aberrant production of these autoantibodies, thereby modifying the endothelial functions to induce cellular immune reactions in vascular walls by acting through vasa vasorum. In fact, the expressions of both EPCR and SR-BI are limited in the vasa vasorum in non-inflammatory aortic tissue and they are stained weaker than in TAK tissue. These data suggest that the

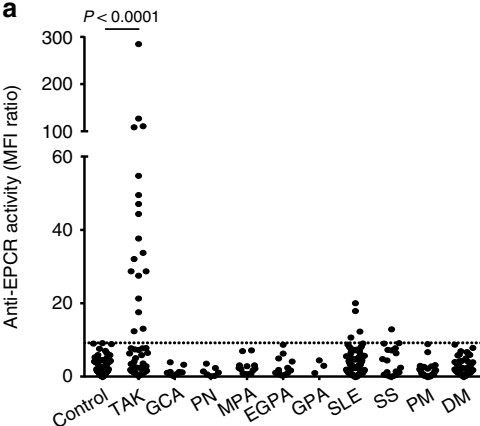

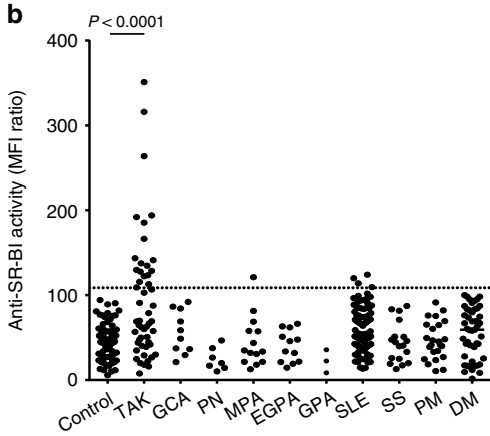

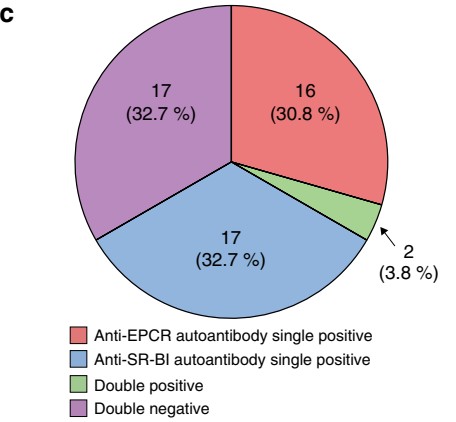

**Fig. 4 Distribution of patients with anti-EPCR and anti-SR-BI activities.**
**a**, **b** The distribution of anti-EPCR (**a**) and anti-SR-BI (**b**) autoantibodies in healthy controls and patients with various autoimmune rheumatic diseases was measured with flow cytometry. Dots represent the data for individual subjects. The broken horizontal line indicates the cut-off value for high activity (mean + 3 SD). Control represents 79 healthy individuals. Autoimmune rheumatic diseases include active Takayasu arteritis (TAK, $n = 52$), giant cell arteritis (GCA, $n = 10$), polyarteritis nodosa (PN, $n = 7$), microscopic polyangiitis (MPA, $n = 14$), eosinophilic granulomatosis with polyangiitis (EGPA, $n = 12$), granulomatosis with polyangiitis (GPA, $n = 3$), systemic lupus erythematosus (SLE, $n = 93$), Sjögren's syndrome (SS, $n = 18$), polymyositis (PM, $n = 24$), and dermatomyositis (DM, $n = 47$). **c** The relative frequency of anti-EPCR and anti-SR-BI autoantibodies in patients with active TAK ($n = 52$) is shown. The numbers of patients and their proportions are indicated in the Figure. Statistical analysis was performed using one-way ANOVA test followed by Tukey's post hoc test. MFI represents mean fluorescent intensity.

**Table 1 Statistical comparison among 50 Takayasu arteritis patients with or without anti-EPCR or anti-SR-BI autoantibodies.**

| | SR-BI+ EPCR− $n = 17$ | SR-BI− EPCR+ $n = 16$ | SR-BI− EPCR− $n = 17$ | P value |
|---|---|---|---|---|
| Stroke | 0 (0%) | 4 (25.0%) | 1 (5.9%) | 0.045 |
| Cardiovascular diseases | 2 (11.8%) | 3 (18.8%) | 4 (23.5%) | 0.67 |
| Aortic regurgitation | 1 (5.9%) | 7 (43.8%) | 8 (47.1%) | 0.017 |
| Ulcerative colitis | 1 (5.9%) | 6 (37.5%) | 0 (0%) | 0.004 |
| Surgical history | 3 (17.7%) | 3 (17.7%) | 7 (41.2%) | 0.19 |
| CRP (mg/dL) | 8.2 ± 7.4 | 3.2 ± 2.9 | 3.3 ± 3.0 | 0.004 |
| ESR (mm/h) | 78.4 ± 35.2 | 42.1 ± 34.2 | 58.1 ± 39.7 | 0.032 |
| Classification of affected vessel | | | | |
| I | 1 (5.9%) | 2 (12.5%) | 2 (11.8%) | 0.78 |
| II | 4 (23.5%) | 10 (62.5%) | 8 (47.1%) | 0.075 |
| V | 11 (64.7%) | 4 (25.0%) | 4 (23.5%) | 0.020 |

Values represent mean ± SD, median (minimum to maximum), or number (%). Categorical variables were analyzed using the $\chi^2$ test. Continuous variables were analyzed using one-way ANOVA.
*CRP* C-reactive protein, *ESR* erythrocyte sedimentation rate.

**Table 2 Positivity of autoantibodies in patients with primary ulcerative colitis.**

| | Positive | Negative | Positivity (%) |
|---|---|---|---|
| Anti-EPCR activity | 24 | 11 | 68.6 |
| Anti-SR-BI activity | 0 | 35 | 0.0 |

expressions of both EPCR and SR-BI are augmented in the inflammatory lesion of TAK. Thus, the mechanisms how they are regulated in vascular inflammation should also be investigated in future.

The limitation of the current study is that the patients were all Japanese and the study population had limited diversity. Although TAK is a rare disease and 80 patients were enrolled in this study, validation in a larger population is required.

## Conclusion

We identified EPCR and SR-BI as endothelial autoantigens in TAK by using SARF. Autoantibodies against these two receptors were specific for TAK among autoimmune rheumatic diseases, and each accounted for one-third of the patients with TAK. Patients with TAK were subclassified on the basis of these autoantibodies, and each subgroup had specific characteristics. Interestingly, anti-EPCR autoantibodies were also detected in patients with primary UC. EPCR and SR-BI ameliorated pro-inflammatory phenotype, and anti-EPCR and anti-SR-BI auto-antibodies promoted pro-inflammatory phenotype by disturbing this negative regulation. Identification of these autoantibodies has opened new avenues for TAK.

## Methods

**Human serum samples**. Three hundred and twenty-five patients with auto-immune rheumatic diseases were enrolled: 80, TAK; 10, GCA; and 235, other autoimmune rheumatic diseases. All the patients were diagnosed according to the respective criteria for classification[42–49]. Analysis flowchart of TAK patients was shown in Supplementary Fig. 1. Disease activity was determined according to National Institutes of Health criteria[50]. Thirty-five patients with UC were also recruited. Seventy-nine age-matched and sex-matched healthy donors were enrolled as the control group. The serum samples obtained from all the subjects were stored at $-20\,^{\circ}\text{C}$ until use. All subjects provided written informed consent after the purpose of and potential risks involved in the study were explained. The study protocol complies with the principles of the Declaration of Helsinki and was approved by the Ethics Committee of Tohoku University Graduate School of Medicine.

**Cell culture**. HUVECs, HAECs, HPAECs, and respective cell culture medium was purchased from Lonza (Basel, Switzerland). Cells were cultured in 5% $CO_2$

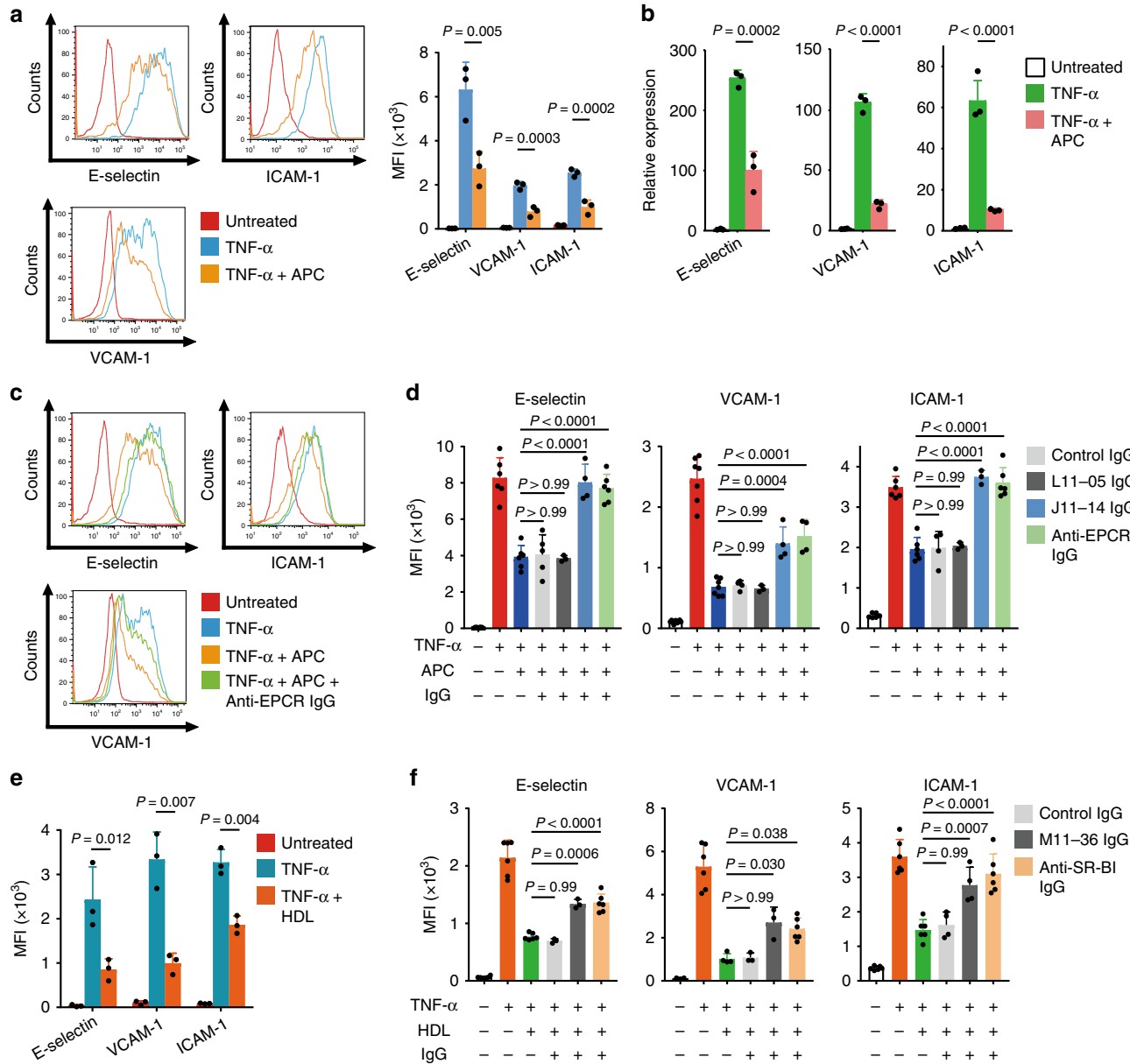

**Fig. 5 Blocking of anti-inflammatory activities of EPCR and SR-BI by autoantibodies in Takayasu arteritis. a, b** HUVECs were treated with or without 10 μg/mL APC for 13 h and stimulated with 100 pg/mL TNF-α for 5 h. The expression of adhesion molecules, including E-selectin, VCAM-1, and ICAM-1, was analyzed with flow cytometry. Representative histograms (left) and the summary graph (right, n = 3) are shown in **a**. The mRNA expression level was measured by quantitative PCR in **b**. GAPDH was used as the internal control. **c** HUVECs were incubated with the isotype or 10 μg/mL anti-EPCR antibody for 1 h and treated with or without 10 μg/mL APC for 13 h. Then, cells were stimulated with 100 pg/mL TNF-α for 5 h. The expression of adhesion molecules was analyzed with flow cytometry. **d** HUVECs were incubated with or without IgG for 1 h. IgG included 10 μg/mL anti-EPCR antibody, 2.56 mg/mL control IgG, 2.56 mg/mL IgG from an AECA-negative TAK sample (L11-05), or 2.56 mg/mL IgG from anti-EPCR-positive TAK AECA sample (J11-14). The cells were subsequently treated as described above, and the expression of adhesion molecules was analyzed. **e** HUVECs were treated with or without 1 mg/mL high-density lipoprotein (HDL) for 16 h and stimulated with 100 pg/mL TNF-α for 5 h. Adhesion molecules were analyzed by flow cytometry; the summary graph is shown (n = 5). **f** HUVECs were incubated with or without IgG for 1 h. IgG included 10 μg/mL anti-SR-BI antibody, 2.56 mg/mL control IgG, or 2.56 mg/mL IgG from anti-SR-BI positive TAK AECAs (M11-36). Cells were subsequently treated with or without 1 mg/mL of HDL as described above, and adhesion molecules were analyzed (n = 3). Data are indicated as mean ± SD. At least three independent experiments were performed in all cases. Statistical analysis was performed using one-way ANOVA followed by Tukey's post hoc test. MFI represents mean fluorescent intensity. L11-05, J11-14, and M11-36 indicate the serum sample number.

at 37 °C and used sooner than the sixth passage. Rat myeloma cells, YB2/0, were purchased from American Type Culture Collection (Manassas, VA, USA) and cultured in RPMI 1640 medium (Sigma-Aldrich, St. Louis, MO, USA) containing 10% fetal bovine serum (FBS, Biowest, Miami, FL, USA). Plat-E packaging cells were purchased from Cell Biolabs (San Diego, CA, USA), and cultured in Dulbecco's Modified Eagle's Medium (DMEM) (Sigma-Aldrich) containing 10% FBS (Biowest). Human peripheral blood mononuclear cells (PBMC) were

isolated by density gradient centrifugation using Ficoll-Paque PLUS (GE Healthcare, Uppsala, Sweden). Naive CD4+ T cells were purified by negative selection using EasySep Human Naive CD4+ T cell Isolation Kit (STEMCELL Technologies, Vancouver, BC, Canada) and were cultured in 96-well U-bottom plates (BD Biosciences, Bedford, MA, USA) in 100 μL RPMI 1640 medium. Adherent cells were dissociated from plates using Cell Dissociation Solution (Sigma-Aldrich).

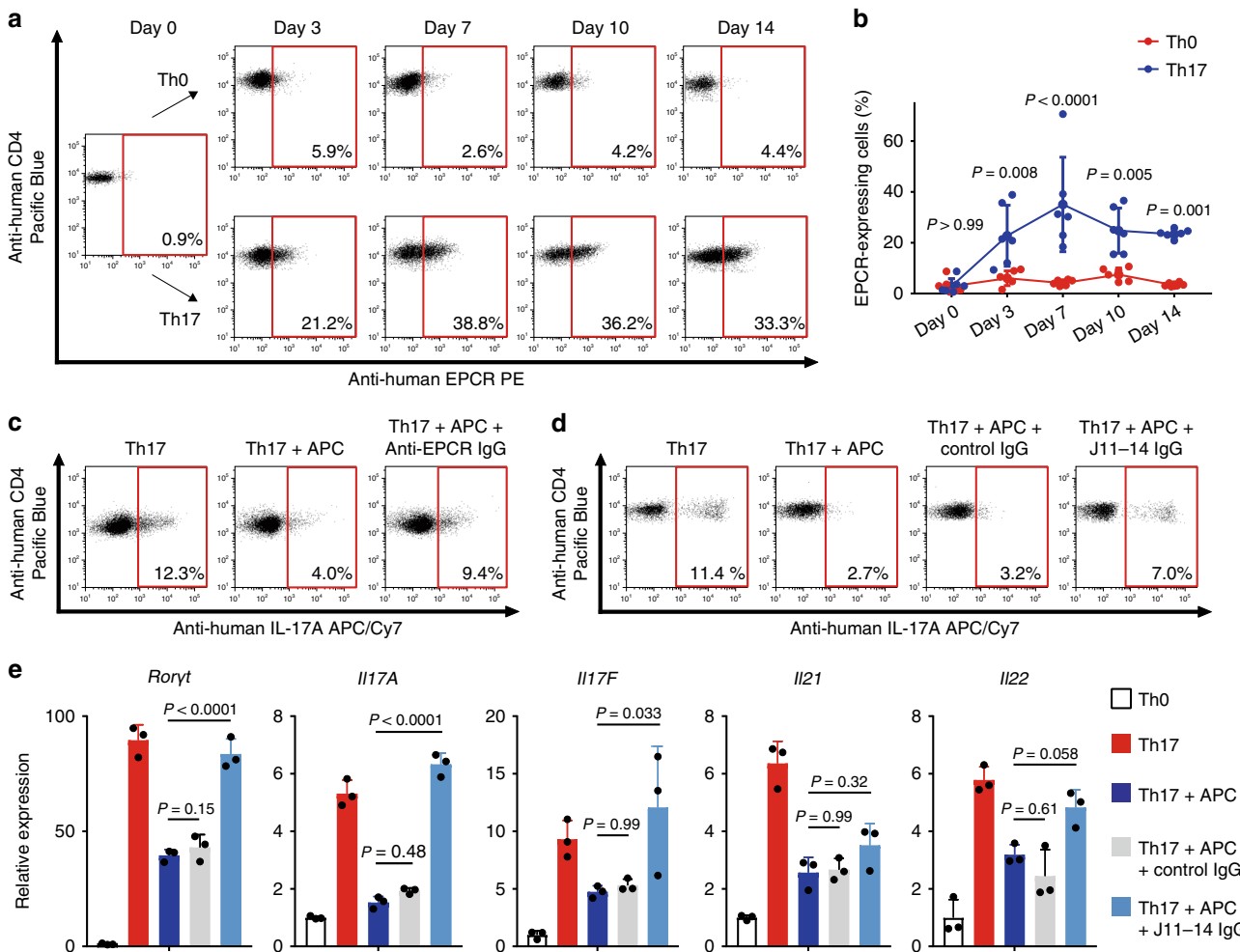

**Fig. 6 Anti-EPCR autoantibodies in Takayasu arteritis negatively regulates Th17 differentiation. a, b** Human naive CD4+ T cells were cultured under Th0 (anti-CD3/anti-CD28 stimulation alone) or Th17 conditions (anti-CD3/anti-CD28 stimulation followed by the addition of interleukin [IL]-6, IL-1β, IL-23, TGF-β1, anti-human interferon-γ antibody, and anti-human IL-4 antibody). The proportion of cells expressing EPCR on day 0, 3, 7, 10, or 14 was analyzed with flow cytometry. Representative dot plots (**a**) and the summary graph (**b** $n = 3$) are shown. **c, d** Human naive CD4+ T cells were stimulated and cultured under Th17 differentiation conditions, and production of IL-17A on Day 7 was analyzed with flow cytometry. Cells were cultured with or without 10 μg/mL APC and incubated with isotype or anti-EPCR antibody (**c**). Cells were cultured with or without 10 μg/mL APC, and control IgG or IgG from an anti-EPCR-positive TAK AECA sample (J11-14) was further added (**d**). **e** Human naive CD4+ T cells were cultured under Th17 conditions with or without IgG and with or without 10 μg/mL APC for 7 days. IgG included 2.56 mg/mL control IgG and 2.56 mg/mL IgG from anti-EPCR positive TAK AECA (J11-14). The relative expression of *Rorγt*, *IL-17A*, *IL-17F*, *IL-21*, and *IL-22* mRNA were measured with quantitative PCR ($n = 3$). GAPDH was used as internal control. Data are indicated as mean ± SD. At least three independent experiments were performed in all cases. Statistical analysis was performed using two-way ANOVA test (**b**) or one-way ANOVA (**e**) followed by Tukey's post hoc test. J11-14 indicates the serum sample number.

**IgG purification.** IgG was purified from sera by using HiTRAP Protein G HP columns (Amersham Biosciences, Roosendaal, The Netherlands) or by Spin column based Antibody Purification Kit (Protein G) (Cosmo Bio, Tokyo, Japan). The concentration of purified IgG was determined by measuring the optical density (OD) at 280 nm. Purified IgG was stored at −20 °C until use.

**Flow cytometry.** Fluorescence intensity was measured using BD LSR Fortessa or FACS Canto II (Becton Dickinson, Franklin Lakes, NJ, USA). Cell sorting was performed using BD FACS Aria II (Becton Dickinson). Dead cells were identified by 7-amino-actinomycin D staining (BD Biosciences) and excluded from analysis. All data was analyzed using FlowJo Software (Tree Star, Ashland, OR, USA). Following antibodies were utilized: FITC-conjugated goat anti-human IgG (Fab specific) (F5512, Sigma-Aldrich, 1:50), PE-conjugated goat F(ab')2 anti-human IgG-Fc (ab98596, Abcam, 1:50), PE-conjugated anti-EPCR antibody (351903, BioLegend, San Diego, CA, USA, 1:50), rat anti-EPCR antibody [RCR-252] (ab81712, Abcam, 1:50), PE-conjugated goat F(ab')2 anti-rat IgG-Fc (ab6259, Abcam, 1:50), PE-conjugated mouse anti-human SR-BI antibody (363203, BioLegend, 1:50), rabbit anti-human SR-BI antibody (PA5-29789, Abcam, 1:50), PE-conjugated goat anti-rabbit IgG (ab72465, Abcam, 1:50), anti-SR-BI antibody (bs-1186R, Bioss antibodies, Woburn, MA, USA, 1:50), rabbit anti-SR-BI antibody

(NB400-113, Novus Biologicals, Littleton, CO, USA, 1:50), PE-conjugated mouse anti-CD62E antibody (336008, BioLegend, 1:100), PE/Cy5-conjugated mouse anti-CD106 antibody (305808, BioLegend, 1:100), PE/Cy7-conjugated mouse anti-CD106 antibody (NBP-47864PECY7, Novus Biologicals, 1:100), Pacific Blue-conjugated mouse anti-CD54 antibody (322716, BioLegend, 1:50), FITC-conjugated mouse anti-human CD3 antibody (555339, BD Biosciences, 1:50), Pacific Blue-conjugated mouse anti-human CD4 antibody (558116, BD Biosciences, 1:50), APC-conjugated mouse anti-human IL-17A antibody (512333, Biolegend, 1:50), APC/Cy7-conjugated mouse anti-human IL-17A antibody (512319, BioLegend, 1:50).

**Measurement of AECA, anti- EPCR, and anti-SR-BI activity.** Binding activity of antibodies in serum to the surface of HUVEC and HAEC was measured by flow cytometry[17,19]. Briefly, 1:10 diluted human serum was used as primary antibody with 50 mg/mL goat gamma globulin fraction (Sigma-Aldrich) and a fluorescent-conjugated antibody was used as secondary antibody. To quantify the activity of AECAs, the relative mean fluorescence intensity (MFI) ratio was determined according to the following calculation formula: (sample MFI—control MFI)/control MFI × 100. Cut-off values for activity of AECAs were determined as the mean + 3 standard deviations (SD) of the relative MFI ratio in control groups. To

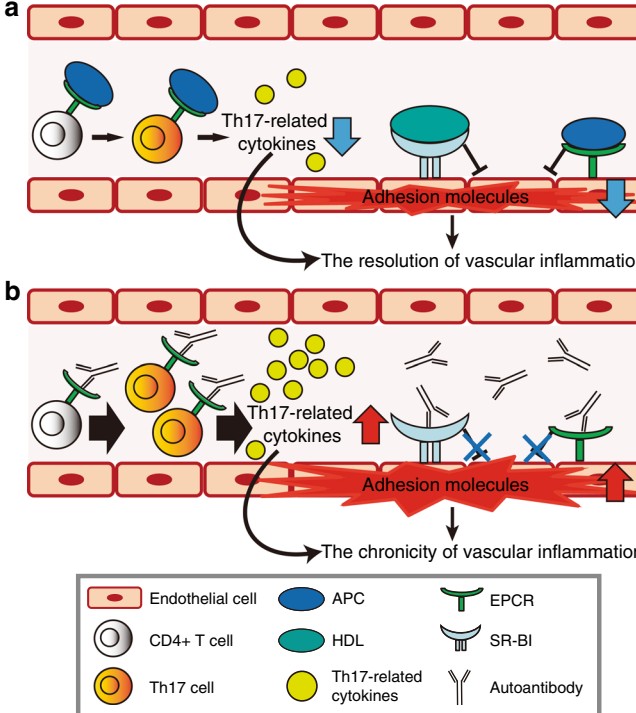

**Fig. 7 Proposed model for the pathogenicity of AECAs in Takayasu arteritis. a** APC and HDL, the ligand of EPCR and SR-BI, respectively, suppress expression of adhesion molecules in activated endothelial cells. APC also negatively regulates the differentiation of CD4+ T cells expressing EPCR into Th17 cells, and thus inhibits the production of Th17-related cytokines, including IL-17A, IL-17F, IL-21, and IL-22. These mechanisms contribute to the resolution of vascular inflammation in normal condition. **b** Autoantibodies against EPCR or SR-BI block bindings of corresponding ligands to their receptors, which promote expression of adhesion molecules in endothelial cells. In addition, anti-EPCR autoantibodies disrupt the negative regulation of Th17 differentiation by APC and thus amplify Th17-mediated immune response. As a result, autoantibodies in TAK inhibit the spontaneous resolution of activated immune responses in the vessel wall and thus lead to the chronic vascular inflammation.

quantify the anti-EPCR or anti-SR-BI activity, the relative MFI ratio was calculated according to the following calculation formula: (MFI of overexpressing cells—MFI of non-overexpressing cells)/MFI of non-overexpressing cells × 100. The cut-off values were determined similarly as mentioned above. The MFI ratio was considered to be the titer of anti-EPCR antibody or anti-SR-BI antibody.

**HUVEC cDNA library**. Two different cDNA libraries were used in this study. One cDNA library was previously generated and has been reported[17,19,20]. New HUVEC cDNA library was generated by using SMART cDNA Library Construction kit (Clontech, Madison, WI, USA), in order to improve efficiency and the generated cDNA was ligated into the pRetro-Lib vector (Clontech).

**cDNA library screening: SARF**. Autoantigen identification was performed using SARF with some modifications[18,20,21]. Briefly, HUVEC cDNA libraries were retrovirally transfected into the YB2/0 rat myeloma cell line (American Type Culture Collection, Manassas, VA, USA). YB2/0 cells expressing the HUVEC cDNA libraries were incubated with serum IgG with high AECA activity and 1:50 diluted fluorescent-conjugated secondary antibody at 4 °C for 30 min. Cells with high fluorescence levels were sorted with FACS Aria II (Becton Dickinson, Franklin Lakes, NJ, USA). The sorted cells were maintained in culture until cell numbers sufficiently increased for the next sorting. Sorting was repeated for several rounds to concentrate cells bound to prototype IgG. Subsequently, several cell clones were established using the limiting dilution method. The genomic DNA of clones was purified, DNA fragments from the HUVEC cDNA libraries were amplified by PCR, and DNA sequences were analyzed with the BLAST program.

**PCR**. DNA fragments from the HUVEC cDNA libraries were amplified by PCR using Takara LA Taq (Takara Bio, Shiga, Japan) with primers corresponding to the 5′ and 3′ ends of the multiple cloning site of pMX (5′-GGTGGACCATCCTCTA GACTG, 3′-CCTTTTTCTGGAGACTAAAT) and pRetro-Lib (5′-AGCCCTCACT CCTTCTCTAG, 3′- ACCTACAGGTGGGGTCTTTCATTCCC) vectors. The PCR products were cloned into pCR-TOPO vector (Invitrogen, San Diego, CA, USA)

**Overexpression of EPCR or SR-BI on YB2/0 cells**. The full-length EPCR and SR-BI fragment were confirmed by DNA sequence analysis of sorted cells as described earlier, by using Phusion High-Fidelity DNA Polymerase (Finnzymes, Keilaranta, Espoo, Finland). Primer sequences were: EPCR (5′-CTAGACTGCCG GATCATGTTGACAACATTGCTGCCG, 3′-CGCGCCGGCCCTCGATTAACAT CGCCGTCCACCTGT) and SR-BI (5′-CTAGACTGCCGGATCAGACATGGGCT GCTCCGC, 3′-CGCGCCGGCCCTCGAGGCTGGCTCACGGTGTCCT). These PCR products were inserted into pMX-IRES-GFP vector (Cell Biolabs) by using In-Fusion Cloning Kit (Clontech). pMX-EPCR-IRES-GFP vector or pMX-SR-BI-IRES-GFP vector were retrovirally transfected into YB2/0 cells.

**Western blotting**. Recombinant EPCR (Abcam) protein was mixed with 2× Laemmli sample buffer (Bio-Rad Laboratories, Hercules, CA, USA). The proteins were electrophoresed on a 12% polyacrylamide gel, and then transferred onto Immobilon transfer membranes (Millipore, Billerica, MA, USA). The membranes were treated with the following antibodies: 1:200 diluted control, U10-4 serum, anti-human EPCR antibody (MAB22451, R&D Systems, Minneapolis, MN, USA, 1:500), IRDye 800CW-conjugated goat anti-human IgG (H + L) (926-32232, LI-COR Biosciences, Lincoln, NE, USA, 1:5000), IRDye 800CW-conjugated goat anti-mouse IgG (926-32210, LI-COR Biosciences, 1:5000). Fluorescence intensity was measured with an Odyssey Infrared Imaging System (LI-COR Biosciences). Uncropped and unprocessed scans were provided in the Source Data file.

**Inhibition test**. Recombinant EPCR (Abcam) or SR-BI protein (R&D Systems) was added at the indicated dose. Inhibition rate (%) was calculated as follows: (AECA activity of sample serum—AECA activity of sample serum with recombinant protein)/AECA activity of sample serum × 100.

**Immunoprecipitation (IP)**. Untransfected YB 2/0 cells and SR-BI transfected YB 2/0 cells were reacted with serum with anti-SR-BI activity or control serum. The cells were lysed with M-PER Mammalian Protein Extraction Reagent (Pierce Biotechnology Inc., Rockford, IL, USA) followed by incubation with 25 μL of Protein A/G Magnetic Beads (Pierce Biotechnology Inc.). After washing with Tris-buffered saline [10 mM tris (pH 7.5), 150 mM NaCl] containing 0.1% Tween 20, eluted with 2× Laemmli sample buffer (Bio-Rad Laboratories). Under non-reducing conditions, Western blotting was performed as described above. The membranes were treated with rabbit anti-SR-BI antibody (NB400-104, Novus Biologicals, 1:1000), IRDye 680-conjugated goat anti-rabbit IgG (926-32221, LI-COR Biosciences, 1:5000), or IRDye 800CW-conjugated goat anti-human IgG (H + L) (926-32232, LI-COR Biosciences, 1:5000).

**Immunohistochemistry**. Formalin-fixed, paraffin-embedded 4-μm-thick sections were deparaffinized. After antigen retrieval by heat/autoclaving 5 min at 121 °C in 10 mM sodium citrate buffer, pH 6.0 for EPCR, SR-BI in EDTA buffer, pH9.0. The sections were incubated with primary antibodies (EPCR 1:200, SR-BI 1:1000) overnight at 4 °C. Primary antibodies included rabbit anti-EPCR antibody (Invitrogen, MA5-29505) and rabbit anti-human SR-BI antibody (Abcam, PA5-29789). Sections were then incubated with EnVision+ system (DAKO) for 30 min. The antigen–antibody complex was visualized with 3,3-diaminobenzidine (DAB) solution (1 mM DAB, 50 mM Tris–HCl buffer [pH 7.6], and 0.006% $H_2O_2$) and counterstained with hematoxylin.

**Analysis of expression of adhesion molecules**. HUVECs were first incubated in 12-well culture plates (BD Biosciences) with IgG for 1 h, followed by the addition of either 10 μg/mL human activated protein C (APC, Hematologic Technologies, Essex Junction, VT, USA) or 1 mg/mL human high density lipoprotein (HDL, Prospec Protein Specialists, Rehovot, Israel). Subsequently, HUVECs were stimulated by 100 pg/mL TNF-α (R&D Systems) for 5 h. Utilized IgG included purified IgG, 10 μg/mL rat anti-human EPCR antibody [RCR-252] (ab81712, Abcam, 1:10), or 10 μg/mL rabbit anti-human SR-BI antibody (PA5-29789, Abcam, 1:100, or NB400-113, Novus Biologicals, 1:100).

**Quantitative reverse transcription polymerase chain reaction (qRT-PCR)**. Total RNA was extracted using the RNeasy Mini Kit (Qiagen, Valencia, CA, USA), and reverse transcribed into cDNA using the ReverTra Ace qPCR reverse transcription (RT) kit (Toyobo, Osaka, Japan). qRT-PCR was performed using QuantiTect SYBR Green PCR master mix (Qiagen) on a C1000 Thermal Cycler (Bio-Rad Laboratories). Gene transcript levels were normalized relative to the levels of GAPDH transcripts. The upstream and downstream PCR primer sequences are indicated as follows: E-selectin: 5′-CAGCAAAGGTACACACACCTG-3′, 5′-CAG ACCCACACATTGTTGACTT-3′

VCAM-1: 5′-TTTGACAGGCTGGAGATAGACT-3′, 5′-TCAATGTGTAATT TAGCTCGGCA-3′

ICAM-1: 5′-ATGCCCAGACATCTGTGTCC-3,′5′-GGGGTCTCTATGCCCA ACAA-3′

Interleukin (IL) -17A: 5′-TCCCACGAAATCCAGGATGC-3′, 5′-GGATGTTC AGGTTGACCATCAC-3′

IL-17F: 5′-GCTGTCGATATTGGGGCTTG-3′, 5′-GGAAACGCGCTGGTTT TCAT-3′

IL-21: 5′-TAGAGACAAACTGTGAGTGGTCA-3′, 5′-GGGCATGTTAGTC TGTGTTTCTG-3′

IL-22: 5′-GCTTGACAAGTCCAACTTCCA-3′, 5′-GCTCACTCATACTGAC TCCGT-3′

RORγT: 5′-CTGCTGAGAAGGACAGGGAG-3′, 5′-AGTTCTGCTGACGGG TGC-3′

GAPDH: 5′-GAAGGTCGGAGTCAACGGATTTT-3′, 5′-GAATTTGCCATGG GTGGAAT-3′.

**Th17 differentiation**. Purified naive CD4+ T cells were cultured with Dynabeads human T-Activator CD3/CD28 (Invitrogen) (1:1 ratio of beads to cells), 10 ng/mL IL-6 (BD Biosciences), 10 ng/mL IL-1β (BD Biosciences), 10 ng/mL IL-23 (R&D Systems), 10 ng/mL TGF-β1 (R&D Systems), 10 µg/mL purified NA/LE mouse anti-human IFN-γ antibody (554699, BD Biosciences, 1:50), and 10 µg/mL purified NA/LE rat anti-human IL-4 antibody (554481, BD Biosciences, 1:100).

**Intracellular staining**. The Th17 cells were stimulated with 50 ng/mL phorbol 12-myristate 13-acetate (PMA, Sigma-Aldrich) and 1 µg/mL ionomycin (Sigma-Aldrich) in the presence of GolgiStop (BD Biosciences) for 6 h at day 7. Cells were fixed with Cytofix/Cytoperm Fixation and Permeabilization Solution Kit (BD Biosciences).

**Statistical analysis**. Statistical analysis was performed using GraphPad Prism 7.03 (GraphPad Software, La Jolla, CA, USA). Categorical variables were analyzed using the $\chi^2$ test or Fisher's exact test. Continuous variables were analyzed using Student's $t$-test or the Mann–Whitney $U$-test, and one-way analysis of variance (ANOVA) or two-way ANOVA followed by Tukey's post hoc tests, or Kruskal–Wallis test followed by the Dunn test was used to compare three or more groups. Pairwise comparisons were analyzed using the paired $t$-test or the Wilcoxon signed-rank test. $P < 0.05$ was considered to be statistically significant. To correct multiplicity in Table 1, $P < 0.005$ was considered to be statistically significant by using Bonferroni method.

**Reporting summary**. Further information on research design is available in the Nature Research Reporting Summary linked to this article.

## Data availability

The data that support the findings of this study are available from the corresponding author upon reasonable request. The source data underlying Figs. 1a, 2a, b, e–h, k, l, 4a–c, 5a, b, d–f, 6b, e and Supplementary Figs. 2c, f, 5a–d, 6a, b, 7, 8a, b, 9, 10, 11b, d, 12b, 13b, 14a, b, 15, 16b, d and 17 are provided as a Source Data file. Microarray data are deposited in the Gene Expression Omnibus (accession code, GSE145367).

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

## Acknowledgements

The authors thank Prof. Masato Nose for technical advice and critical reading of this manuscript. We also thank Ms. Yayoi Aoyama (Tohoku University Hospital) for technical assistance for immunohistochemistry and Ms. Chikako Fushimi for technical assistance for flow cytometry and Western blotting. This work was supported by JSPS KAKENHI Grant Numbers 16H06642 and 18K16136 for T.S. and research grants from the Kanae Foundation and Nanbyo Medical Foundation for T.S.

## Author contributions

T.M. measured titers of autoantibodies and performed in vitro experiments. T.S. performed SARF, identified autoantigens, and performed in vitro experiments. T.M. and T.S. analyzed data. T.S., Y.S., T.I., Y.K., Y.K., A.M., Y.S., H.H., and H.F. enrolled patients. T.S. and F.F. evaluated immunohistochemistry. T.S. and F.T. analyzed clinical characteristics. T.S. and H.F. conceived the study, and designed experiments. T.S. wrote the manuscript.

## Competing interests

The authors declare no competing interests.
