## [Peer Review File · Nature Communications]

Reviewers' comments:

Reviewer #1 (Remarks to the Author):

The manuscript by Mutoh and colleagues identify EPCR and SR-B1 as endothelial surface antigens for autoimmune antibodies that are generated in patients with Takayasu arthritis. Upon identification of EPCR and SR-B1, various experiments are performed to validate the recognition of these targets by the patient's sera and to demonstrate that these patients' antibodies interfere with normal functions of these endothelial cell surface proteins. Overall, the manuscript is comprehensive in the studies performed but the analyses are primarily qualitative, using FACS analyses, and there are very limited quantitative analyses performed. Also, the results are mainly descriptive and while several experiments were performed to demonstrate that the antibodies inhibited EPCR and SR-B1 function, this was done in the presence of very high concentrations of added ligand, such as APC, and antibody >2 mg/ml. The question remains what the effects are of these antibodies in more physiological systems, and to what extent are these antibodies generated primary or secondary to development of the disease.

- In many panels essential controls are missing. E.g., in Fig 1 the expression levels of EPCR and SR-B1 in untransfected YB2/O cells. In Fig 4 the effect of high concentration anti-sera in the absence of APC or HDL.

- Fig 1F does not contribute anything. It is well known that HUVEC's express EPCR. Supp Fig 12 no antibody included, changes in gene expression profiles by APC on endothelial cells have been published many years ago.

- Why were only few clones analyzed after the limited dilution of the selected pool? Did other clones show different bands?

- Some descriptions are very difficult to follow without added information. E.g., line 126: microarray analysis showed that EPCR and SR-B1 signals increased in cloned cells.

- Line 136/137 The anti-EPCR or anti-SR-B1 activity of these serum samples was confirmed by inhibition tests (suppl Fig 3). It seems however that only a single patient sample was analyzed and not all positive antibody samples.

- Some of the IgG concentrations required to neutralize the effects of APC seem very high (>1 mg/ml IgG). Without a titer determination and some insight into how much anti EPCR-antibody is present in these IgG fraction, it is difficult to determine the relevance of these experiments for more physiological systems.

Reviewer #2 (Remarks to the Author):

Mutoh T & Shirai T et al investigate the presence and specificity of anti-endothelial cell antibodies (AECA) in sera from patient with Takayasu arteritis (TAK). By using a serological identification system for autoantigens using a retroviral vector and flow cytometry (SARF) technique, developed by T Shirai (co-first author) the authors identify 2 major specificities for AECA in TAK: endothelial protein C receptor (EPCR) and scavenger receptor class B type 1 (SR-BI). Autoantibodies recognizing these proteins are present in 34.6% and 36.5% respectively of a cohort of 52 patients with TAK. These autoantibodies are exceptionally found in the sera from patients with other vasculitis or autoimmune diseases analyzed. The presence of these autoantibodies is rarely coincident and their presence is associated with particular clinical or angiographic features in patients with TAK. The auto-antibodies against EPCR and SR-BI block the inhibitory activity of these proteins on endothelial cell activation in vitro. Moreover EPCR may be expressed by cells other than endothelial cells (i.e. stimulated T lymphocytes) and the auto-antibodies interfere with EPCR inhibitory activity on Th17 differentiation. The authors hypothesize that, through these mechanisms, autoantibodies anti-EPCR and anti-SR-BI may facilitate vascular inflammation. The authors are cautious about attributing to these antibodies a primary pathogenic role but postulate a role in amplifying vascular inflammation and a role as diagnostic and activity assessment tool in patients with TAK.

While many aspects of this manuscript are interesting, there are some methodological issues that need to be addressed.

- 1) The authors use an interesting technique they have developed but wide validation by other investigators is not available. Binding specificity of the autoantibodies identified should be confirmed by western-blot with appropriate controls.
- 2) All experiments are performed using HUVEC cells. It would be necessary to show how EPCR and SR-BI are expressed in target TAK tissue and whether autoantibodies from patients bind to TAK tissue.
- 3) The authors remark the suitability of their method to identify cell surface antigens. However, flow cytometry is performed in cells in suspension. Endothelial cells are naturally adherent and polarized cells and release from substrate may lead to changes and internalization of relevant surface antigens. The authors remark also the potential pathogenic relevance of cell surface antigens as compared with intracellular antigens but this statement is not entirely accurate: anti-neutrophil cytoplasmic antibodies (ANCA), with demonstrated pathogenicity, recognize intracellular antigens. Intracellular antigens may be exposed by damaged cells or can be translocated to the surface under a variety of stimuli.
- 4) It is not specified how the TAK patient cohort was assembled, how diagnosis was supported and how disease activity was assessed. In the abstract it is stated that 80 patients with TAK were studied but in the text it appears that autoantibodies were measured in 52 patients.
- 5) The authors claim that detecting these autoantibodies may be clinically useful in terms of diagnosis and disease activity assessment. However the individual sensitivity is low and specificity should be more robustly validated.

6) It is interesting that specificities of the auto-antibodies investigated are almost mutually exclusive in patients with TAK. Based on this, the authors hypothesize that they may be associated with different phenotypes. However, the number of patients analyzed is small for subgrouping and no correction for multiple comparisons is applied. Therefore, conclusions are weakly supported by data.

7) Given the association of SR-BI specificity with concomitant ulcerative colitis found by the authors, disease controls should include patients with primary ulcerative colitis. Among disease controls, the authors include patients with giant-cell arteritis. It would be relevant to know whether or not these patients had large-vessel involvement. It would be also important to include among disease controls patients with severe atherosclerosis and thrombotic diseases.

Minor

- EPCR and SR-BI need to be spelled in the title.

Reviewer #3 (Remarks to the Author):

In a variety of autoimmune disorders, the presence of autoantibodies that recognize endothelial cells (AECA) is well known; however, the identification of the antigens that may contribute to pathogenesis has been challenging. To specifically find the cell surface antigens, the authors' group has developed a method called SARF, which combines expression cloning system with flow cytometry. Using SARF, the group has already reported multiple antigens in SLE, rheumatoid arthritis and necrotizing encephalopathy. In the current work, Mutoh T. et al. identified two receptors, EPCR and SR-B1 as autoantigens for AECA in Takayasu arteritis (TAK).

These proteins were thoroughly validated as TAK specific autoantigens, and importantly the plasma concentrations of these proteins are uniquely associated with clinical characteristics or diseases status.

However, there are a few concerns;

1) SR-B1 plays a critical role in cardiovascular health by regulating circulating lipid profiles in the liver as a part of the reverse cholesterol transport. In humans, a loss-of-function variant of SR-B1 profoundly affects plasma levels of high-density cholesterol (HDL) (Science. 351:1166, 2016). If the autoantibody inhibits HDL actions in endothelial cells (as demonstrated in Fig. 4), it may also affect them in the liver, considering its effect on HDL uptake (Sup Fig. 8). Thus, the plasma lipid profiles,

such as LDL- and HDL- cholesterol levels, in the patients in the study need to be included. The correlation of the lipid status can be analyzed against the presence or absence of SR-B1 antibody, along with vascular disease-related parameters (as shown in Table 1).

2) Functional testing of the antibodies was performed using HUVEC (Fig. 4). It is well established that the endothelial cells from different vascular beds behave differently. The inhibitory effect of the autoantibody needs to be tested in more pertinent endothelial cells, such as aortic, coronary or pulmonary endothelial cells.

3) HDL exerts anti-inflammatory actions mainly by increasing NO bioavailability. The process does not require internalization of HDL (J Lipid Res,54:2315, 2013). Because the antibody inhibits uptake of HDL to HUVEC (Sup. Fig. 8), it is important to demonstrate whether suppression of eNOS activation underlies the inhibitory action of the SR-B1 antibody.

We really thank the reviewers for their thorough and careful review. Their suggestions were helpful and led to new points of view. Responses to the reviewers are described below.

Response to Reviewer #1:

The manuscript by Mutoh and colleagues identify EPCR and SR-B1 as endothelial surface antigens for autoimmune antibodies that are generated in patients with Takayasu arthritis. Upon identification of EPCR and SR-B1, various experiments are performed to validate the recognition of these targets by the patient's sera and to demonstrate that these patients' antibodies interfere with normal functions of these endothelial cell surface proteins. Overall, the manuscript is comprehensive in the studies performed but the analyses are primarily qualitative, using FACS analyses, and there are very limited quantitative analyses performed. Also, the results are mainly descriptive and while several experiments were performed to demonstrate that the antibodies inhibited EPCR and SR-B1 function, this was done in the presence of very high concentrations of added ligand, such as APC, and antibody >2 mg/ml. The question remains what the effects are of these antibodies in more physiological systems, and to what extent are these antibodies generated primary or secondary to development of the disease.

- 1) *In many panels essential controls are missing. E.g., in Fig 1 the expression levels of EPCR and SR-B1 in untransfected YB2/0 cells. In Fig 4 the effect of high concentration anti-sera in the absence of APC or HDL.*

We had performed these experiments and presented data in the new Supple Fig 2 and the new Supple Fig 7a.

- 2) *Fig 1F does not contribute anything. It is well known that HUVEC's express EPCR. Supp Fig 12 no antibody included, changes in gene expression profiles by APC on endothelial cells have been published many years ago.*

We presented these data to confirm our findings and improve understanding of readers. According to the suggestion from reviewer #1, we removed these data in this revision.

- 3) Why were only few clones analyzed after the limited dilution of the selected pool? Did other clones show different bands?

We established 12 clones from each samples, and analyzed their inserted cDNA as shown below. Most of the cDNA inserted were intracellular molecules, and we searched for the molecules expressed on the cell surface. Because IgG-bound cells were efficiently concentrated in U10-4 and W10-59 as shown in Fig. 1, about half of clones possessed cDNA of PROCR (U10-4) or SCARB1 (W10-59), respectively. On the other hand, it was not easy to concentrate bound cells in G10-43 because of its weaker AECA activity than previous two sera, most clones contained only cDNA of the intracellular molecules and SCARB1 was inserted in the C7 clone. To investigate whether other membrane proteins can be inserted into cloned cells, we performed microarray analysis to compare mRNA between cloned cells and untransfected YB2/0 cells. As shown in Supple Fig 3, the expressions of PROCR and SCARB1 were markedly enhanced, supporting the PCR data.

PCR of cDNA inserted into clones.

U10-4

W10-59

G10-43

- 4) *Some descriptions are very difficult to follow without added information. E.g., line 126: microarray analysis showed that EPCR and SR-B1 signals increased in cloned cells.*

Thank you for the comment. Because of the word limit, we had shortened sentences, and it would have made it difficult to follow the manuscript. We added following sentence in Result section,

“To further confirm cDNA inserted into cloned cells, we performed microarray analysis to compare expressions of mRNA between cloned cells and untransfected YB2/0 cells.”

- 5) *Line 136/137 The anti-EPCR or anti-SR-B1 activity of these serum samples was confirmed by inhibition tests (suppl Fig 3). It seems however that only a single patient sample was analyzed and not all positive antibody samples.*

We had performed inhibition tests for other samples, and presented data in the new Supple Fig 4.

- 6) *Some of the IgG concentrations required to neutralize the effects of APC seem very high (>1 mg/ml IgG). Without a titer determination and some insight into how much anti EPCR-antibody is present in these IgG fraction, it is difficult to determine the relevance of these experiments for more physiological systems.*

We speculate concerns from the reviewer #1 about the concentrations of antibodies probably because we used IgG at the concentration of 2.56 mg/mL and such high doses are not usually used in experiments with mice and in vitro studies. The normal level of IgG in human serum is about 15 mg/mL, and that of TAK patients is 25-30 mg/mL because they manifest hypergammaglobulinemia. Therefore, the concentration of IgG used in our study corresponds to that of the 1:10 diluted serum. Regarding experiments investigating autoantibodies in human, 1:10 diluted serum has been applied to determine pathogenic roles (please refer Arthritis Rheum. 2006 Jul;54(7):2326-33., Kidney Int. 2005 Aug;68(2):537-41., Eur J Clin Invest. 2014 Aug;44(8):753-65.). In the initial submission, we had presented data about experiments with titration of IgG (Supple Fig 7 and 13). The concentration of IgG in physiological condition is 10 times higher and it is more concentrated in the inflammatory lesion.

To address this concern, we performed experiments using commercially available antibodies which we used in the manuscript, and we converted the concentration of patient

total IgG to the corresponding concentration of EPCR/SR-BI-specific antibodies. As a result, 2.56 mg/dL of J11-14 IgG manifested anti-EPCR activity corresponding to 6.5 $\mu\text{g/mL}$ of commercial anti-EPCR antibody, 10 $\mu\text{g/mL}$ of which showed significant blocking effects as shown in New Fig.5. Similarly, 2.56 mg/dL of M11-36 IgG manifested anti-SR-BI activity corresponding to 7.3 $\mu\text{g/mL}$ of commercial anti-SR-BI antibody (New Fig.5). Because the actual concentration of IgG is 10 times higher, it is reasonable to consider that these autoantibodies play pathogenic roles in vivo. We hope these answers appropriately addressed concerns from the reviewer. We revised the Result section as follows,

“Pre-incubation of HUVECs with anti-EPCR antibody or AECA IgG with anti-EPCR activity (J11-14) at the concentration corresponding to 1:10 diluted serum blocked the protective effect of APC on adhesion molecules (Fig. 5c, d),”

Dose-response curve of anti-EPCR antibody against the expression of E-selectin on HUVEC

Response to Reviewer #2:

Mutoh T & Shirai T et al investigate the presence and specificity of anti-endothelial cell antibodies (AECA) in sera from patient with Takayasu arteritis (TAK). By using a serological identification system for autoantigens using a retroviral vector and flow cytometry (SARF) technique, developed by T Shirai (co-first author) the authors identify 2 major specificities for AECA in TAK: endothelial protein C receptor (EPCR) and scavenger receptor class B type 1 (SR-BI). Autoantibodies recognizing these proteins are present in 34.6% and 36.5% respectively of a cohort of 52 patients with TAK. These autoantibodies are exceptionally found in the sera from patients with other vasculitis or autoimmune diseases analyzed. The presence of these autoantibodies is rarely coincident and their presence is associated with particular clinical or angiographic features in patients with TAK. The auto-antibodies against EPCR and SR-BI block the inhibitory activity of these proteins on endothelial cell activation in vitro. Moreover EPCR may be expressed by cells other than endothelial cells (i.e. stimulated T lymphocytes) and the auto-antibodies interfere with EPCR inhibitory activity on Th17 differentiation. The authors hypothesize that, through these mechanisms, autoantibodies anti-EPCR and anti -SR-BI may facilitate vascular inflammation. The authors are cautious about attributing to these antibodies a primary pathogenic role but postulate a role in amplifying vascular inflammation and a role as diagnostic and activity assessment tool in patients with TAK.

While many aspects of this manuscript are interesting, there are some methodological issues that need to be addressed.

- 1) The authors use an interesting technique they have developed but wide validation by other investigators is not available. Binding specificity of the autoantibodies identified should be confirmed by western-blot with appropriate controls.

This was a tough question because not all antibodies can be used in every application including flow cytometry, immunohistochemistry, and Western blotting. Particularly, binding activity of antibodies is influenced by the spatial structure and post-translational modifications of their target membrane proteins. Important examples of autoantibodies which can only be detected by cell-based assay are anti-MOG autoantibody and anti-AQP4 autoantibody in encephalitis (please refer *Neurology*. 2019 Mar 12;92(11):e1250-e1255, *Lancet*. 2004 Dec 11-17;364(9451):2106-12.). In these diseases, autoantigens are membrane proteins and cell-based assay is required for their detection. In other words, these autoantigens were not able to be identified by the methods which change the

conformation of native autoantigens.

Furthermore, extraction of some membrane proteins is difficult in proteomics approach. These are the reasons why we developed SARF to identify cell-surface autoantigens which seemed to be difficult to be detected in conventional methods.

As used in anti-MOG encephalitis (Neurology. 2019 Mar 12;92(11):e1250-e1255), cell based assay which we used in this manuscript is a widely accepted detection method. To confirm the specificity of autoantibodies, we further performed inhibition assay using recombinant protein and the binding activities of autoantibodies were inhibited, which confirmed their specificity to the target protein.

To address this concern, we first performed Western blotting of the cells over-expressing EPCR or SR-BI. We confirmed that these proteins are overexpressed in transfected cells (new Supple Fig 2), but extraction of EPCR was difficult and it was detected as a smear. Reaction with the prototype TAK-AECA was not able to show their binding activities to proteins extracted from overexpressed cells, and one reason for EPCR was considered to be the limit of sensitivity. We then performed Western blotting using recombinant proteins and investigated the binding activity of TAK-AECA. As a result, we were able to confirm the reaction of prototype U10-4 serum to the recombinant EPCR protein (new Fig 2f). However, the detection of binding activity using immunoblotting was more difficult than using cell-based assay when we compared the contrast of the band detected. As for SR-BI, we were not able to detect binding activity of W10-59 or G10-43 serum against recombinant protein in Western blotting. Therefore, we next reacted W10-59 serum with SR-BI-over-expressing cells, lysed cells, and performed immunoprecipitation followed by Western blotting in various conditions including reducing and non-reducing conditions. As a result, we were able to confirm the binding activity of W10-59 and G10-43 to SR-BI. However, this method was complicated and difficult to perform compared with cell-based assay. Especially for SR-BI, spatial structure was important for the detection by anti-SR-BI autoantibody. Because binding activities of AECA to the cells were inhibited by the addition of recombinant EPCR/SR-BI (new Supple Fig 4), we believe that it was convincing that they bound specifically to EPCR/SR-BI.

We revised the Result section as follows,

“In addition, the binding activity of U10-4 serum to recombinant EPCR protein was confirmed by Western blotting (Fig. 2f).”

“Importantly, anti-SR-BI activity of AECA was not documented by the standard Western

blotting. However, we confirmed anti-SR-BI activity by using immunoprecipitation (Fig. 2I), indicating that the spatial structure of SR-BI protein was important for the binding of anti-SR-BI autoantibodies.”

2) *All experiments are performed using HUVEC cells. It would be necessary to show how EPCR and SR-BI are expressed in target TAK tissue and whether autoantibodies from patients bind to TAK tissue.*

We performed immunohistochemistry using resected aorta from TAK patients and made new Fig. 3. Interestingly, EPCR and SR-BI are both expressed in the endothelial cells of the affected aortic lumen, and also intensely found in the endothelium of vasa vasorum. The main inflammatory site of TAK is in vasa vasorum, and thickening of the intimal layers has been considered as the secondary phenomenon although TAK is classified as large vessel vasculitis (Ann Vasc Surg. 2016 Aug;35:210-25.). New Fig. 3 supported the importance of EPCR and SR-BI in the regulation of vasa vasorum vasculitis, a feature of TAK.

We also performed staining of IgG in TAK tissue, which suggested the deposition of IgG in the endothelium of vasa vasorum as shown below. However, we are reluctant to make definite conclusion from the immunohistochemistry for human IgG because staining of human IgG in human tissue sometimes manifest non-specific staining.

Immunohistochemistry of TAK tissue with anti-human IgG

We made new Fig.3 and added following sentences in Result and Discussion section,

Results

“Expression of EPCR and SR-BI in TAK tissue

Although thickening of intimal layers of the aorta is the hallmark of TAK, this has been

considered as the secondary phenomenon and the main inflammatory site of TAK is in vasa vasorum²⁴. To investigate the expression of autoantigens in TAK, immunohistochemistry was performed using resected aorta from TAK patients. Vasa vasorum vasculitis with infiltration of inflammatory cells was observed in TAK tissue, and endothelium of vasa vasorum expressed both EPCR and SR-BI (Fig 3). EPCR and SR-BI were also expressed in the endothelial cells of the affected aortic lumen, but their expressions were more intense in the vasa vasorum, suggesting their roles in vasa vasorum vasculitis.”

Discussion

“One hypothesis is that vascular damage accompanied by immune activation initiates and leads to aberrant production of these autoantibodies, thereby modifying the endothelial functions to induce cellular immune reactions in vascular walls by acting through vasa vasorum.”

3) *The authors remark the suitability of their method to identify cell surface antigens. However, flow cytometry is performed in cells in suspension. Endothelial cells are naturally adherent and polarized cells and release from substrate may lead to changes and internalization of relevant surface antigens. The authors remark also the potential pathogenic relevance of cell surface antigens as compared with intracellular antigens but this statement is not entirely accurate: anti-neutrophil cytoplasmic antibodies (ANCA), with demonstrated pathogenicity, recognize intracellular antigens. Intracellular antigens may be exposed by damaged cells or can be translocated to the surface under a variety of stimuli.*

We really appreciate the comment from the reviewer #2. We also agree that each method has advantage and disadvantage, and should be used complementarily. We also understand that autoantibodies against intracellular molecules play critical roles in the pathogenicity as is the case for ANCA-associated vasculitis. Because it was possible that these points were emphasized, we revised some parts of manuscript according to the comment from reviewer #2.

Introduction

“Target antigens of AECAs are heterogeneous and include membrane component, ligand-receptor complex, and molecule adhering to plasma membrane¹⁶. Autoantigens may be either constitutively expressed or translocated from intracellular compartment to membrane. Anti-neutrophil cytoplasmic antibodies are one of the autoantibodies which

recognize intracellular antigens and their pathogenic roles have been implicated¹⁷”

Discussion

“Meanwhile, endothelial cells are naturally adherent and polarized cells. Cells should be detached from the plate for detection by flow cytometry, and this may lead to change and internalization of surface antigens. Taken together, use of SARF for autoantigen identification could be considered for other diseases in which AECAs have been reported¹⁸.”

4) *It is not specified how the TAK patient cohort was assembled, how diagnosis was supported and how disease activity was assessed. In the abstract it is stated that 80 patients with TAK were studied but in the text it appears that autoantibodies were measured in 52 patients.*

This study consisted of discovery phase and validation phase, and each phase contained 21 patients and 59 patients, respectively. Patients satisfied ACR or JCS classification criteria for TAK. In validation phase, 7 patients were excluded because they were considered inactive according to the NIH criteria and 52 patients were examined. We presented this flow in new Supple Fig 1.

5) *It is interesting that specificities of the auto-antibodies investigated are almost mutually exclusive in patients with TAK. Based on this, the authors hypothesize that they may be associated with different phenotypes. However, the number of patients analyzed is small for subgrouping and no correction for multiple comparisons is applied. Therefore, conclusions are weakly supported by data.*

Thanks for your suggestion. We discussed about this matter with a statistician, Dr. F.T. who was included as a co-author. Because the number of patients analyzed was small as pointed out by the reviewer, we decided to select 10 parameters which we considered as clinically important and performed statistical analysis as shown in new Table 1. Then, $P < 0.005$ was considered to be statistically significant to correct multiplicity using Bonferroni method. As a result, complication of UC and CRP levels were significantly different. Although we were not able to conclude significant, we described about other parameters for the possibility for subgrouping. We are now planning a multicenter-prospective study about these autoantibodies in large vessel vasculitis, and hopefully show the further validation in future.

We revised manuscripts as follows,

Result section

“Their clinical characteristics have been presented in Supplementary Table 1, 2. We selected 10 parameters which we considered as clinically important and performed statistical analysis as shown in Table 1. To correct multiplicity, $P < 0.005$ was considered to be statistically significant. Anti-SR-BI-positive patients were relatively older (mean, 41.2 years), and aortic regurgitation (AR) was relatively less than in other types. They exhibited elevated levels of inflammatory markers, including C-reactive protein ($P = 0.004$), and relative elevation of erythrocyte sedimentation rate; 64.7% patients had type V artery lesions. Patients with anti-EPCR autoantibodies tended to experience more strokes (25.0%) and had significantly higher frequencies of ulcerative colitis (UC) ($P = 0.004$). Lesser numbers of arteries were affected, and 62.5% patients had type II artery lesions. Patients without these autoantibodies had increased rates of surgery (41.2%), most of which were performed for AR.”

Discussion section

“Because disease-specific markers are extremely helpful, further validation in a larger population should be conducted.”

Method section

“To correct multiplicity in Table 1, $P < 0.005$ was considered to be statistically significant by using Bonferroni method.”

6) *The authors claim that detecting these autoantibodies may be clinically useful in terms of diagnosis and disease activity assessment. However, the individual sensitivity is low and specificity should be more robustly validated.*

Given the association of SR-BI specificity with concomitant ulcerative colitis found by the authors, disease controls should include patients with primary ulcerative colitis. Among disease controls, the authors include patients with giant-cell arteritis. It would be relevant to know whether or not these patients had large-vessel involvement. It would be also important to include among disease controls patients with severe atherosclerosis and thrombotic diseases.

To evaluate the specificity, we have increased the number of patients with other collagen diseases as possible as we could and total number of patients with collagen disease

became 325. However, we also agree that this point is one of the limitation of this study, which we described in the Discussion section. We further collaborated with the division of gastroenterology, and measured anti-EPCR autoantibodies in patients with the primary ulcerative colitis (UC). Unexpectedly, we found that approximately 70 % of patients with primary UC possessed anti-EPCR activities (new Table 2). This result was not only very surprising but also impressive because anti-EPCR activities were specific for TAK among collagen diseases and autoantibodies with such high prevalence have not known in UC. Nonetheless, this result strongly supported our result in the first submission that the complication of UC was one of the features in anti-EPCR positive TAK. UC is the most prevalent comorbidity in TAK, and these two diseases share similar comorbidities including spondyloarthritis and pyoderma gangrenosum. Taken together, our results further revealed the common pathogenesis among TAK and UC, and it is possible that these diseases can be categorized by the presence of anti-EPCR autoantibodies which possess pathogenic roles. In other words, these two diseases might be one syndrome with different organ manifestation. This finding is very exciting and should also have significant impact on the field of medicine. We added following sentences in Result section and discussed about them in Discussion section.

Results

“Anti-EPCR activity was detected in 1 of 18 patients with Sjögren’s syndrome and 4 of 93 patients with systemic lupus erythematosus (SLE); anti-SR-BI activity was detected in 1 of 14 patients with microscopic polyangiitis and 4 of 93 patients with SLE. In summary, the sensitivity and specificity of two autoantibodies in collagen diseases were 67.3% and 98.0%, respectively.”

“Because the complication of UC was significantly higher in patients with anti-EPCR autoantibodies, we measured anti-EPCR autoantibodies in 35 patients with primary UC. Surprisingly, 68.6 % of UC sera possessed binding activities to EPCR (Table 2, Supplementary Fig. 6). Anti-SR-BI autoantibodies were not detected in primary UC.”

Discussion

“On the other hand, we further revealed that patients with primary UC possessed autoantibodies against EPCR. Crucial role of EPCR in governing microvascular inflammation in inflammatory bowel disease has been reported³⁶. This data in itself has significant impact on the research for UC because of its high prevalence. The co-existence of TAK and UC has been known, and they further share common complications such as

spondyloarthritis and pyoderma gangrenosum³⁷. Therefore, this result suggests the similar pathogenesis among TAK and UC, which can be characterized by the presence of anti-EPCR autoantibodies. Further studies are required to evaluate association among TAK and UC based on this autoantibody.”

Regarding GCA, three patients manifested large-vessel involvement and we described it in the Result section.

“Ten patients who had giant cell arteritis with positive results in temporal artery biopsy possessed neither of them although three patients manifested large-vessel involvement.”

We also agree with the reviewer that the measurement in atherosclerosis and thrombotic diseases would be important. However, this study protocol included only healthy controls and patients with autoimmune diseases. Because the results of this study generated various important outcomes which should also be investigated in other vascular diseases, we discussed about them in Discussion section and would like to investigate them in future research.

“However, the inhibition of SR-BI by autoantibodies could be the risk for developing atherosclerotic lesions whose occurrence is known to be high in TAK⁴⁰. Further investigations of these autoantibodies in severe atherosclerosis and thrombotic disease would also be important.”

Minor

1) *EPCR and SR-BI need to be spelled in the title.*

We revised the title according to the suggestion.

“Protein C receptor and scavenger receptor class B type 1 negatively regulate vascular inflammation and represents novel endothelial autoantigens in Takayasu arteritis”

Response to Reviewer #3:

In a variety of autoimmune disorders, the presence of autoantibodies that recognize endothelial cells (AECA) is well known; however, the identification of the antigens that may contribute to pathogenesis has been challenging. To specifically find the cell surface antigens, the authors' group has developed a method called SARF, which combines expression cloning system with flow cytometry. Using SARF, the group has already reported multiple antigens in SLE, rheumatoid arthritis and necrotizing encephalopathy. In the current work, Mutoh T. et al. identified two receptors, EPCR and SR-B1 as autoantigens for AECA in Takayasu arteritis (TAK).

These proteins were thoroughly validated as TAK specific autoantigens, and importantly the plasma concentrations of these proteins are uniquely associated with clinical characteristics or diseases status.

However, there are a few concerns;

- 1) SR-B1 plays a critical role in cardiovascular health by regulating circulating lipid profiles in the liver as a part of the reverse cholesterol transport. In humans, a loss-of-function variant of SR-B1 profoundly affects plasma levels of high-density cholesterol (HDL) (Science. 351:1166, 2016). If the autoantibody inhibits HDL actions in endothelial cells (as demonstrated in Fig. 4), it may also affect them in the liver, considering its effect on HDL uptake (Sup Fig. 8). Thus, the plasma lipid profiles, such as LDL- and HDL-cholesterol levels, in the patients in the study need to be included. The correlation of the lipid status can be analyzed against the presence or absence of SR-B1 antibody, along with vascular disease-related parameters (as shown in Table 1).*

We evaluated the plasma lipid profiles including total cholesterol, LDL-cholesterol, HDL-cholesterol, and triglycerides levels in TAK patients as shown in Supplementary table 1, and investigated the correlation between the lipid status and anti-SR-BI antibodies. There did not exist significant difference in the lipid profiles depending on the presence of autoantibodies in our cohort. Because anti-SR-BI positive patients manifested higher inflammatory activities compared with other subpopulations, it is possible that the difference in the inflammatory status also influenced the lipid profiles. We added following sentences in Discussion section.

“In humans, a loss-of-function variant of SR-BI was reported to be correlated with increased levels of HDL, and an increased risk of coronary heart disease³⁹. In our cohort, there did not

exist significant difference in the lipid profiles depending on the presence of autoantibodies. It is possible that lipid profiles were affected by the higher inflammatory activities of anti-SR-BI positive subjects. However, the inhibition of SR-BI by autoantibodies could be the risk for developing atherosclerotic lesions whose occurrence is known to be high in TAK⁴⁰. Further investigations of these autoantibodies in severe atherosclerosis and thrombotic disease would also be important.”

2) *Functional testing of the antibodies was performed using HUVEC (Fig. 4). It is well established that the endothelial cells from different vascular beds behave differently. The inhibitory effect of the autoantibody needs to be tested in more pertinent endothelial cells, such as aortic, coronary or pulmonary endothelial cells.*

As shown in the new Fig 3, the main inflammatory site of TAK is in vasa vasorum, and thickening of the intimal layers has been considered as the secondary phenomenon although TAK is classified as large vessel vasculitis (Ann Vasc Surg. 2016 Aug;35:210-25.). Therefore, it has been speculated that the actual target of the inflammation including anti-endothelial cell antibody would be against vasa vasorum. Actually, EPCR and SR-BI are both expressed in the endothelial cells of the affected aortic lumen, and also intensely found in the vasa vasorum. These data confirmed the importance of autoantibodies against them in the vasa vasorum vasculitis, a feature of TAK. However, it is difficult to perform experiments with endothelial cells from vasa vasorum because of its availability. According to the suggestions from the reviewer #3, we performed validation in different endothelial cells using human aortic endothelial cells and pulmonary artery endothelial cells. As a result, similar findings were observed and presented in Supple Fig 10 and 15.

We revised the Result section as follows,

“Other endothelial cells including HAECs and human pulmonary endothelial cells (HPAECs) also expressed EPCR, and similar blocking effect of anti-EPCR autoantibodies was also observed in these cells (Supplementary Fig. 10).”

“HAECs and HPAECs also expressed SR-BI, and anti-SR-BI autoantibody blocked the effects of HDL (Supplementary Fig. 15).”

3) *HDL exerts anti-inflammatory actions mainly by increasing NO bioavailability. The process does not require internalization of HDL (J Lipid Res,54:2315, 2013). Because the antibody inhibits uptake of HDL to HUVEC (Sup. Fig. 8), it is important to demonstrate whether suppression of eNOS activation underlies the inhibitory action of*

the SR-B1 antibody.

Thank you for your important advice. We examined eNOS activation using OxiSelect™ intracellular Nitric Oxide (NO) assay kit, and made new Supple Fig 16. eNOS activation was induced by the addition of HDL, and anti-SR-BI autoantibody inhibited the activation of eNOS. Therefore, it was possible that eNOS activation underlid the inhibitory action of the SR-BI antibody.

We added following sentences in Result section.

“HDL exerts anti-inflammatory actions mainly by increasing nitric oxide bioavailability³¹. Measurement of nitric oxide synthase (NOS) activity revealed that suppression of NOS activation underlid the inhibitory action of anti-SR-BI autoantibody (Supplementary Fig. 16).”

Reviewers' comments:

Reviewer #1 (Remarks to the Author):

The authors have adequately responded to the concerns noted. However, the official name for EPCR is ENDOTHELIAL protein C receptor, and the word endothelial needs to be added to title, key words, and first use of EPCR in the abstract.

Reviewer #2 (Remarks to the Author):

T Mutoh and collaborators present a revised and clearly improved version of their manuscript.

They have appropriately addressed the reviewer's questions and have added a significant amount of experimental work which strengthens the scientific content of their work

However, there is still an opportunity of improvement. Specific suggestions are the following:

1) In general the authors should not make strong statements about aspects that are not fully demonstrated (see also comment 7). For example, the title states that protein receptor C and scavenger receptor class B type 1 negatively regulate vascular inflammation. The authors show that the autoantibodies inhibit the negative effects of these proteins on the expression of pro-inflammatory molecules and other functional activities relevant to inflammation in cultured endothelial cells (quite interesting) but, strictly speaking, the experiments performed do not demonstrate an effect on vascular inflammation (i. e. as in animal model for example). The impact on vascular inflammation may be likely but remains hypothetical. The title should be more focused on the identification of novel endothelial autoantigens in Takayasu patients and the functional impact of the autoantibodies against these antigens on endothelial cell inflammatory phenotype in vitro. At the end of the abstract it is also stated that the autoantibodies promote vascular inflammation. It should be changed to endothelial activation or pro-inflammatory phenotype.

2) New immunohistochemistry data demonstrating expression of the autoantigens in endothelial cells from arteries involved by Takayasu are relevant. As a comparator, immunohistochemistry on normal aortas or large vessels should be provided. These antigens may or may not be expressed in normal arteries and this may be important to know.

3) Throughout the text the authors mix the potential diagnostic role of the autoantibodies identified with their potential usefulness as biomarkers of disease activity. These are different roles that not always run in parallel. It needs to be clarified to what of these roles the authors refer in each statement. It is likely that the autoantibodies identified by the authors may not be excellent diagnostic biomarkers individually since their sensitivity is low but detection of either one may have

higher sensitivity Their specificity is much better although anti-EPCR antibodies can also be detected in ulcerative colitis (very interesting finding!).

4) In the introduction the authors mention (line 61) that the presence of anti-aortic antibodies have been documented. Could the authors be more precise about whether these antibodies recognize endothelial cells, vascular smooth muscle cells or structural components?

5) In the methods section, the source of the antibodies used for immunohistochemistry is not depicted.

6) In lines 215- -226, the authors describe interesting experiments demonstrating that anti SR-BI autoantibodies from Takayasu inhibit HDL uptake by endothelial cells and that commercially obtained autoantibodies against SR-BI (I am not sure that commercial antibodies are really autoantibodies) blocked the protective effect of HDL on TNF-activated endothelial cells. Why Takayasu autoantibodies were not used for these functional studies?. It would have been interesting also to demonstrate that this was specific for anti SR-BI and that anti EPCR did not elicit this effect.

7) The authors show that anti SR-BI antibodies decrease NOS activity. Since HDL is known to exert anti-inflammatory actions by decreasing NO bioavailability the authors conclude that suppression of NOS activation underlies the inhibitory action of anti SR-BI autoantibodies. There is indeed no experimental demonstration for this mechanism. It should be discussed as an interesting possibility, not as a result since the data generated does not demonstrate this point. The authors show that antibodies decrease NOS activation and decrease HDL intake but do not demonstrate a link between both observations

Minor

1) I am really aware of the tremendous effort that the authors have invested in writing in a foreign language but the manuscript should be revised in depth for the accuracy of some statements

In line 44 (abstract) the authors say that autoantibodies against EPCR or SR-BI accounted for 34.6% or 36.5 % of cases. Perhaps it would be better to say that autoantibodies against EPCR or SR-BI were detected in 34.6 % and 36.5 % of cases respectively.

In line 48 (abstract) the authors refer to mechanical studies. They probably mean mechanistic or functional studies

In line 60, the authors say: In contrast to such cellular immune reactions, B cells... 'In addition' would be more accurate than 'in contrast'.

In line 69-70, the authors state that autoantigens may be constitutively expressed 'or' translocated from intracellular compartment to membrane. These are not antagonist situations. For example, ANCA autoantigens are constitutively expressed 'and' translocated to membrane.

In line 78 something is missing between the period and the following sentence.

The authors frequently refer to collagen diseases when discussing about autoimmune or immune-mediated diseases. Currently, autoimmune diseases is a more accurate name than collagen diseases which may refer to conditions derived from genetic/structural abnormalities in the collagen molecules

2) In the introduction the authors mention that there are a few animal models of Takayasu. I am not aware of any animal model of this disease. I am aware of certain mice that develop large-vessel vasculitis (i.e. IL-1RA KO mice, IRF-4-binding protein KO mice, virus infected interferon-deficient mice etc). If the authors are aware of an animal model of Takayasu, references need to be added.

Reviewer #3 (Remarks to the Author):

The revised manuscript has thoroughly addressed my previous concerns.

We again thank reviewers for thorough review and kind suggestions. We truly believe their suggestions significantly improved the quality of our manuscript. Responses to the reviewers are described below.

Response to Reviewer #1

The authors have adequately responded to the concerns noted. However, the official name for EPCR is ENDOTHELIAL protein C receptor, and the word endothelial needs to be added to title, key words, and first use of EPCR in the abstract.

We revised the name of EPCR throughout the manuscript. We appreciate your careful review.

Response to Reviewer #2

T Mutoh and collaborators present a revised and clearly improved version of their manuscript.

They have appropriately addressed the reviewer's questions and have added a significant amount of experimental work which strengthens the scientific content of their work

However, there is still an opportunity of improvement. Specific suggestions are the following:

1) In general the authors should not make strong statements about aspects that are not fully demonstrated (see also comment 7). For example, the title states that protein receptor C and scavenger receptor class B type 1 negatively regulate vascular inflammation. The authors show that the autoantibodies inhibit the negative effects of these proteins on the expression of pro-inflammatory molecules and other functional activities relevant to inflammation in cultured endothelial cells (quite interesting) but, strictly speaking, the experiments performed do not demonstrate an effect on vascular inflammation (i. e. as in animal model for example). The impact on vascular inflammation may be likely but remains hypothetical. The title should be more focused on the identification of novel endothelial autoantigens in Takayasu patients and the functional impact of the autoantibodies against these antigens on endothelial cell inflammatory phenotype in vitro. At the end of the abstract it is also stated that the autoantibodies promote vascular inflammation. It should be changed to endothelial activation or pro-inflammatory phenotype.

Thank you for the comment. We revised the title as follows,

“Endothelial protein C receptor and scavenger receptor class B type 1 negatively regulate **endothelial activation** and represent novel autoantigens in Takayasu arteritis”

Also, we revised some sentences as follows,

Abstract

“Autoantibodies against EPCR and SR-BI blocked the functions of their targets, thereby promoting **pro-inflammatory phenotype**.”

Results

“Thus, anti-SR-BI autoantibodies also promoted **endothelial activation**.”

Conclusion

“EPCR and SR-BI ameliorated pro-inflammatory phenotype, and anti-EPCR and anti-SR-BI autoantibodies promoted pro-inflammatory phenotype by disturbing this negative regulation.”

2) New immunohistochemistry data demonstrating expression of the autoantigens in endothelial cells from arteries involved by Takayasu are relevant. As a comparator, immunohistochemistry on normal aortas or large vessels should be provided. These antigens may or may not be expressed in normal arteries and this may be important to know.

The problem we faced was that it was difficult to get normal aorta. Therefore, we performed immunohistochemistry in non-inflammatory aortic tissue. These included surgical specimen from patients with aortic aneurysm and aortic stenosis, and autopsy samples from patients with hypothyroidism and amniotic fluid embolism. The expression of EPCR was not evident in the intima, and the endothelium of vasa vasorum was stained weakly. The expression of SR-BI was also detected in the endothelium of vasa vasorum, whereas the intimal layer was not stained in some samples. These data suggested that the expressions of both EPCR and SR-BI were augmented in the inflammatory lesion of TAK. We made new Supple Fig 4 for these images, and added following sentences in Result section,

“Immunohistochemistry was also performed in non-inflammatory aortic tissue to investigate expressions of EPCR and SR-BI. Surgical specimen from patients with aortic aneurysm and aortic stenosis, and autopsy samples from patients with hypothyroidism and amniotic fluid embolism were analyzed. The expression of EPCR was not evident in the intima, and the endothelium of vasa vasorum was stained weakly in non-inflammatory aortic tissue (Supplementary Fig. 4a). The expression of SR-BI was also detected in the endothelium of vasa vasorum, whereas the intimal layer was not stained in some samples of non-inflammatory aortic tissue (Supplementary Fig. 4b).”

in Discussion section,

“In fact, the expressions of both EPCR and SR-BI were limited in the vasa vasorum in non-inflammatory aortic tissue and they were stained weaker than in TAK tissue. These data suggested that the expressions of both EPCR and SR-BI were augmented in the inflammatory lesion of TAK. Thus, the mechanisms how they are regulated in vascular inflammation should also be investigated in future.”

3) Throughout the text the authors mix the potential diagnostic role of the autoantibodies

identified with their potential usefulness as biomarkers of disease activity. These are different roles that not always run in parallel. It needs to be clarified to what of these roles the authors refer in each statement. It is likely that the autoantibodies identified by the authors may not be excellent diagnostic biomarkers individually since their sensitivity is low but detection of either one may have higher sensitivity Their specificity is much better although anti-EPCR antibodies can also be detected in ulcerative colitis (very interesting finding!).

I agree with the reviewer that there exist several possibilities for the clinical application including diagnosis, subclassification, and monitoring disease activity.

We added following sentences in each statement in Discussion section.

“These data suggested the potential diagnostic role of these autoantibodies.”

“These data suggested the potential use of these autoantibodies for the subclassification of TAK.”

“The data from serial measurements suggest that these autoantibodies reflect disease activity, suggesting their roles for monitoring disease activity. Taken together, these autoantibodies could be used as a diagnostic, subclassification, and monitoring tool in clinical practice.”

4) In the introduction the authors mention (line 61) that the presence of anti-aortic antibodies have been documented. Could the authors be more precise about whether these antibodies recognize endothelial cells, vascular smooth muscle cells or structural components?

Initially, homogenized human aorta was used as an antigen. Complement fixation test, haemagglutination test were initially performed and the ELISA was then used for detection. We revised the sentence as follows,

“Initially, the presence of antiaortic antibodies was documented by complement fixation test and hemagglutination test with homogenized human aorta^{7,8}”

5) In the methods section, the source of the antibodies used for immunohistochemistry is not depicted.

We added the information.

“Primary antibodies included rabbit anti-EPCR antibody (Invitrogen, MA5-29505) and rabbit anti-human SR-BI antibody (Abcam, PA5-29789).”

6) In lines 215- -226, the authors describe interesting experiments demonstrating that anti SR-BI autoantibodies from Takayasu inhibit HDL uptake by endothelial cells and that commercially obtained autoantibodies against SR-BI (I am not sure that commercial antibodies are really autoantibodies) blocked the protective effect of HDL on TNF-activated endothelial cells. Why Takayasu autoantibodies were not used for these functional studies?. It would have been interesting also to demonstrate that this was specific for anti SR-BI and that anti EPCR did not elicit this effect.

In this experiment, we used “commercially available anti-SR-BI IgG (Supple Fig12)” and “anti-SR-BI autoantibodies: M11-36 (Supple Fig13)”. It seemed the sentence was difficult to understand, we revised it as follows,

“Commercially available anti-SR-BI IgG and anti-SR-BI-positive TAK AECAs also dose-dependently blocked the protective effect of HDL on TNF- α -treated HUVECs (Fig. 5f, Supplementary Fig. 12-13),”

Anti-EPCR autoantibodies did not interfere the uptake of HDL, which was presented as new Supple Fig 12c. We added following sentence in Result section,

“Anti-EPCR-positive TAK AECAs did not interfere the uptake of HDL (Supplementary Fig. 12c)”

7) The authors show that anti SR-BI antibodies decrease NOS activity. Since HDL is known to exert anti-inflammatory actions by decreasing NO bioavailability the authors conclude that suppression of NOS activation underlies the inhibitory action of anti SR-BI autoantibodies. There is indeed no experimental demonstration for this mechanism. It should be discussed as an interesting possibility, not as a result since the data generated does not demonstrate this point. The authors show that antibodies decrease NOS activation and decrease HDL intake but do not demonstrate a link between both observations

We revised the sentence as follows,

“Measurement of nitric oxide synthase (NOS) activity revealed that anti-SR-BI autoantibodies suppressed NOS activity (Supplementary Fig. 16).”

We moved following sentence to Discussion.

“Particularly, it was possible that suppression of NOS activation underlid the inhibitory action of anti-SR-BI autoantibodies.”

Minor

1) *I am really aware of the tremendous effort that the authors have invested in writing in a foreign language but the manuscript should be revised in depth for the accuracy of some statements*

Thank you for pointing these issues. We revised sentences according to the suggestion from reviewer #2.

In line 44 (abstract) the authors say that autoantibodies against EPCR or SR-BI accounted for 34.6% or 36.5 % of cases. Perhaps it would be better to say that autoantibodies against EPCR or SR-BI were detected in 34.6 % and 36.5 % of cases respectively.

We revised it as indicated.

“Autoantibodies against EPCR or SR-BI were detected in 34.6% and 36.5% of cases, respectively, with minimal overlap (3.8%).”

In line 48 (abstract) the authors refer to mechanical studies. They probably mean mechanistic or functional studies

We revised it as indicated.

“In mechanistic studies,”

In line 60, the authors say: In contrast to such cellular immune reactions, B cells... ‘In addition’ would be more accurate than ‘in contrast’.

We revised it as indicated.

“In addition to such cellular immune reactions”

In line 69-70, the authors state that autoantigens may be constitutively expressed ‘or’ translocated from intracellular compartment to membrane. These are not antagonist situations. For example, ANCA autoantigens are constitutively expressed ‘and’ translocated

to membrane.

We revised it as indicated.

“Autoantigens may be either constitutively expressed and translocated”

In line 78 something is missing between the period and the following sentence.

Here, we used “ : ” to introduce SARF. It seemed that the position of the citation made it difficult to distinguish “ : ” from “ . ”. Therefore we moved the citation.

“Therefore, we constructed a novel expression cloning system to identify cell-surface antigens: serological identification system for autoantigens using a retroviral vector and flow cytometry (SARF)^{18,20}”

The authors frequently refer to collagen diseases when discussing about autoimmune or immune-mediated diseases. Currently, autoimmune diseases is a more accurate name than collagen diseases which may refer to conditions derived from genetic/structural abnormalities in the collagen molecules

Thank you for the comment. We would like to distinguish autoimmune rheumatic diseases from autoimmune diseases which include organ-specific autoimmune diseases such as inflammatory bowel diseases. Therefore, we revised “collagen disease” to “autoimmune rheumatic diseases”.

2) In the introduction the authors mention that there are a few animal models of Takayasu. I am not aware of any animal model of this disease. I am aware of certain mice that develop large-vessel vasculitis (i.e. IL-1RA KO mice, IRF-4-binding protein KO mice, virus infected interferon-deficient mice etc). If the authors are aware of an animal model of Takayasu, references need to be added.

In this sentence, we intended to mention about the mouse model of large vessel vasculitis which reviewer #2 listed up. As pointed out by the reviewer, they are the animal models of large vessel vasculitis and not of Takayasu arteritis to be exact. We revised this sentence as follows,

“Because there does not exist an animal model, TAK studies have been conducted using human samples; its pathogenesis is mostly unclear^{2,3}”

Response to Reviewer #3

The revised manuscript has thoroughly addressed my previous concerns.

We really appreciate your review.

REVIEWERS' COMMENTS:

Reviewer #2 (Remarks to the Author):

The authors satisfactorily answered most of my comments.

-Data supporting the association between autoantibody specificity and clinical phenotypes is not satisfactorily robust due to the limited number of patients in each group and the borderline statistical significance after correction for multiple comparisons (there is no mention, indeed, to what specific methods for correction the authors used). The authors have appropriately softened the conclusions about these findings in the main text. However, mention to this potential association is still included in the abstract among the major findings of the paper. Since data are not strong enough, the following sentence: `Patients with TAK and different types of autoantibodies showed distinct clinical characteristics`) should be removed from the abstract

- New data added about expression of EPCR and SR-BI in non-inflammatory aortic tissue (control tissue) is important and would be better included as additional panels in figure 3 (currently showing expression of EPCR and SRBI in Takayasu tissue only) rather than including them as supplementary data. The reader will better appreciate the differences. Panels can be made smaller or the figure just include one picture of expression in vasa vasorum per patient to make space for controls.

Minor:

In the abstract, the authors mention that EPCR functioned in human T cells and ameliorated Th17 differentiation. I suggest modifying the sentence as follows: EPCR had also an effect on human T cells and impaired Th17 differentiation.

Maria C Cid, MD

We again thank reviewers for thorough review. Responses to the reviewer are described below.

Response to Reviewer #2

The authors satisfactorily answered most of my comments.

-Data supporting the association between autoantibody specificity and clinical phenotypes is not satisfactorily robust due to the limited number of patients in each group and the borderline statistical significance after correction for multiple comparisons (there is no mention, indeed, to what specific methods for correction the authors used). The authors have appropriately softened the conclusions about these findings in the main text. However, mention to this potential association is still included in the abstract among the major findings of the paper. Since data are not strong enough, the following sentence: `Patients with TAK and different types of autoantibodies showed distinct clinical characteristics`) should be removed from the abstract

We had corrected multiplicity using Bonferroni method, and had mentioned it in the Method section in previous submission. We included the method in Result section as well.

“To correct multiplicity, $P < 0.005$ was considered to be statistically significant using Bonferroni method.”

Regarding abstract, we removed the sentence as suggested by the reviewer.

- New data added about expression of EPCR and SR-BI in non-inflammatory aortic tissue (control tissue) is important and would be better included as additional panels in figure 3 (currently showing expression of EPCR and SRBI in Takayasu tissue only) rather than including them as supplementary data. The reader will better appreciate the differences. Panels can be made smaller or the figure just include one picture of expression in vasa vasorum per patient to make space for controls.

We revised Figure 3 and included IHC from controls. Expressions in the intima are shown in Supple Fig. 4.

Minor:

In the abstract, the authors mention that EPCR functioned in human T cells and ameliorated Th17 differentiation. I suggest modifying the sentence as follows: EPCR had also an effect on human T cells and impaired Th17 differentiation.

We revised the sentence as suggested by the reviewer 2.